# Mining Hidden Thoughts from Texts: Evaluating Continual Pretraining with Synthetic Data for LLM Reasoning

## Abstract

Large Language Models (LLMs) have demonstrated significant improvements in reasoning capabilities through supervised fine-tuning and reinforcement learning. However, when training reasoning models, these approaches are primarily applicable to specific domains such as mathematics and programming, which imposes fundamental constraints on the breadth and scalability of training data. In contrast, continual pretraining (CPT) offers the advantage of not requiring task-specific signals. Nevertheless, how to effectively synthesize training data for reasoning and how such data affect a wide range of domains remain largely unexplored. This study provides a detailed evaluation of Reasoning CPT, a form of CPT that uses synthetic data to generate the **hidden thought** processes underlying texts, based on the premise that texts are the result of the author's thinking process. Our analysis shows that Reasoning CPT can significantly enhance reasoning ability even when trained on non-STEM corpora that have rarely been used for reasoning tasks. On both MMLU and GPQA, Reasoning CPT achieved substantial improvements over the base model and standard CPT. For instance, on GPQA Diamond, performance improved from 23.7% with the base model to 32.8% with Reasoning CPT, while on MMLU the benefits became more pronounced as problem difficulty increased, with gains of up to 11.2 points on the hardest questions. Most notably, models trained with hidden thoughts from legal texts outperformed models trained with standard CPT on STEM data, strongly suggesting that reasoning abilities can be enhanced not only from STEM corpora but also from diverse domains, opening a new direction beyond the conventional STEM-centric paradigm of reasoning model training.

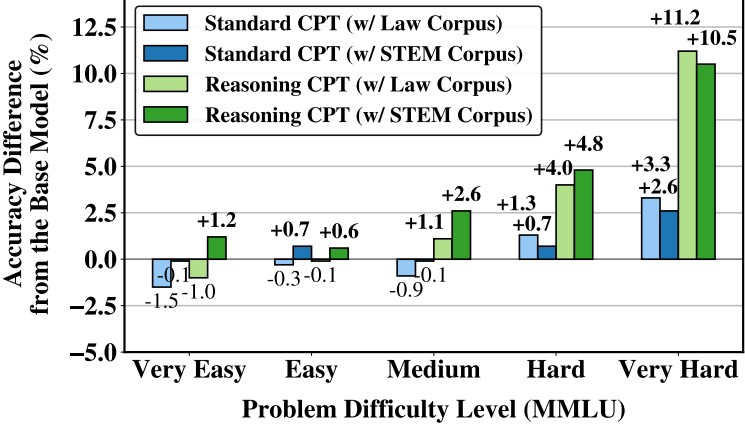

Figure 1: Performance differences from the base model across MMLU problem difficulty levels

## 1 INTRODUCTION

Improving the reasoning capabilities of large language models (LLMs) is a central research challenge. Recent inference-time scaling methods—exemplified by OpenAI o1 (Jaech et al., 2024) and DeepSeek R1 (DeepSeek-AI et al., 2025)—now reach human-expert performance on complex tasks such as the AIME (of America, 2024), programming contests, and advanced scientific reasoning (Rein et al., 2023).

Previous research (Jaech et al., 2024; DeepSeek-AI et al., 2025; Muennighoff et al., 2025) has mostly used instruction tuning (SFT) (Muennighoff et al., 2025) and reinforcement learning (RL) (Shao et al., 2024) to improve reasoning capabilities in domains with clear correctness criteria, such as mathematics and programming. Despite their recent successes, RL- and SFT-based training regimes aimed at boosting LLM reasoning share a fundamental bottleneck: both rely on explicit, task-specific reward signals. This constraint keeps them effective only in reward-rich domains such as mathematics and programming, preventing them from exploiting the abundant, high-quality text available in other fields.

In contrast, pretraining does not depend on specific domains or explicit reward signals. The challenge lies in constructing suitable training data for reasoning. A natural approach is to mine thinking processes from texts and use them as training data for continual pretraining. The key insight here is that every text can be seen as the result of an author's implicit thought processes. For example, mathematical proof texts show logical flow, but behind them are many decisions involving trial and error, hypothesis verification, and consideration of counterexamples. Similarly, legal opinions reflect logical conclusions concisely, but judges recall precedents and consider legal interpretations and social impacts from multiple perspectives. We refer to these hidden internal thought processes as *hidden thoughts*.

Generating hidden thoughts behind texts with LLMs provides a promising pathway for creating extensive reasoning data across diverse domains. Unlike traditional methods such as SFT or RL, this approach has several potential benefits: (1) it does not require strict correctness checks; (2) it efficiently leverages existing high-quality texts; and (3) it enables training reasoning abilities across various domains. Despite these advantages, the potential of continual pretraining with hidden thought processes remains insufficiently explored.

In this study, we evaluate the effect of incorporating hidden thoughts into continual pretraining across two different domains: STEM and Law. We use LLMs to simulate human hidden thoughts involved in creating expert-quality texts—such as recalling background knowledge, making decisions, verifying steps, and expressing thoughts naturally—to build synthetic datasets. Specifically, we generate hidden thoughts using `Gemma2-9B-it` (Shao et al., 2024) for specialized texts from two sources: OpenWebMath (Paster et al., 2024) and FreeLaw (Gao et al., 2021), and combine them with the original texts. We compare two continual pretraining settings applied to base models: one using the original texts (CPT), and another using synthetic data that combines original texts with hidden thoughts (*Reasoning CPT*). Our evaluation on the MMLU benchmark reveals several key findings:

- **Cross-Domain Transfer of Reasoning Capabilities**: Reasoning CPT consistently outperforms standard CPT across all test domains, achieving performance improvements of up to 3.3 points over the base model. A particularly interesting finding is that significant improvements are observed even in domains different from those used for training. For example, models trained with hidden thoughts from the law domain show improvements not only in MMLU social sciences but also in MMLU-STEM domains by 4.3 points.

- **Higher Performance on Difficult Problems:** The advantages of Reasoning CPT become more pronounced as problem difficulty increases. For the most challenging problems, it achieves accuracy improvements of approximately 8 points compared to standard CPT (Figure 1).

- **Non-STEM Training Surpassing STEM Models:** Remarkably, Reasoning CPT trained on non-STEM texts (Law domain) achieved higher accuracy on STEM benchmarks than models trained on STEM texts with standard CPT. On GPQA, the law-trained Reasoning CPT with Qwen2.5-7B reached 30.3% accuracy, exceeding the 25.3% of STEM-trained CPT. This result highlights that reasoning skills acquired from non-STEM texts can transfer effectively to scientific tasks, even surpassing domain-specific training.

## 2 SYNTHETIC DATA CONSTRUCTION AND CONTINUAL PRETRAINING

This study explores how continual pretraining with implicit thinking processes (hidden thoughts) extracted from domain-specific expert texts affects LLMs' reasoning capabilities. We evaluate this approach in two different domains—STEM and Law—by generating hidden thoughts and measuring its impact. This section explains how we collect data, create synthetic data with hidden thoughts, and perform continual pretraining.

### 2.1 DATA COLLECTION

We collect texts from STEM and Law domains for continual pretraining. These texts serve two purposes: as training data for standard CPT and as source material for generating hidden thoughts.

**STEM Domain**   We collect texts from OpenWebMath[1] (Paster et al., 2024), which contains forum posts, educational content, reference pages, scientific papers, and blogs from Common Crawl. This dataset includes texts about mathematics, physics, computer science, statistics, chemistry, and other STEM fields. We preprocess the data through the following steps. First, we limit each example to a maximum of 512 tokens for efficient training, filtering out longer texts. Then, we use `Gemma2-9B-it` (Shao et al., 2024) to remove low-quality content such as chat dialogues, emails, dates, URLs, and non-English text. We process the texts to fit within the 512-token limit and cut them at proper sentence endings to maintain readability. From this cleaned dataset, we randomly selected 150,000 examples, totaling 36.8M tokens.

**Law Domain**   Legal opinions are well-suited for training LLM reasoning skills because they contain complex legal reasoning based on extensive knowledge. We extract U.S. legal case documents from the FreeLaw subset of The Pile (Gao et al., 2021). We focus on the opinion sections, looking for parts that begin with "JUSTICE," "CHIEF JUSTICE," or "PER CURIAM," or that contain phrases such as "The issue in this case" or "We granted certiorari," which typically start legal reasoning. For texts without these markers, we randomly select paragraphs. We keep text length between 64 and 512 tokens and cut them at sentence endings to maintain readability. We also remove footnote numbers (like "[1]", "[2]") and page numbers to improve text quality. We only keep formally structured paragraphs that begin with capital letters. From this processed dataset, we randomly select 150,000 examples, totaling 28.3M tokens.

### 2.2 SYNTHETIC DATA CONSTRUCTION

We create synthetic data by adding LLM-generated hidden thoughts to the original STEM and legal texts described in §2.1. We use this synthetic data for Reasoning CPT, while standard CPT uses the original texts without hidden thoughts. This setup helps us clearly measure the impact of hidden thoughts on model performance, as the only difference between standard CPT and Reasoning CPT is the presence of hidden thoughts, with the same underlying texts.

**Hidden Thought**   Expert-written texts are the results of underlying thought processes that are not explicitly shown. For example, a mathematical proof is the outcome of a mathematician trying multiple approaches, drawing on background knowledge, and conducting verification, while the final text only presents the conclusions. We focus not only on background knowledge recall but also on the human decision-making and reasoning processes behind text creation. Within the framework of this study, we assume that texts are the products of human reasoning processes and use LLMs to generate these implicit thought trajectories. We then construct synthetic datasets by combining such generated hidden thoughts (unexpressed reasoning behind the text) with the original texts.

Figure 2 shows an example of our synthetic data. As shown in this example, the hidden thoughts we generate include four characteristic aspects that capture how domain experts typically think:

1. **Human-like thinking style**: Spontaneous expressions such as "Hmm..." or "Aha!" (e.g.,"Hmm, what kind of information are we talking about?")

---

[1]`https://huggingface.co/datasets/open-web-math/open-web-math`

2. **Background knowledge recall**: Internally recalling relevant rules, concepts, or facts mentioned (or presupposed) in the text (e.g., "Let me recall the specific terms of the protective order. It protects sensitive information such as price proposals...")

3. **Decision-making**: Considering an action (e.g., "I need to justify redacting information from the court decision.")

4. **Self-verification**: Checking the reasoning for possible mistakes or omissions (e.g., "Let me double-check if there are any exceptions to the protective order that might apply here.")

---

```
<start_of_thought>
```
**OK, I need to justify redacting information from the court decision.**
First, what's the basis for redaction? It's likely a protective order. I need to remember the specific terms of that order. Right, it protects sensitive information.
**Wait, what kind of information are we talking about?** Proposals, prices, and price discussions... this sounds like commercially sensitive data.
**Hmm, I could argue that this information is confidential under the protective order,** but that's a bit generic. Let me think about the specific harms of disclosure.
**Aha! Releasing this information would give Torres and G4S's competitors an unfair advantage.** They could see how Torres and G4S approach procurements and potentially undercut their bids. That's a clear competitive disadvantage.
**I should focus on this competitive harm argument.** It's more specific and directly addresses the nature of the information being redacted.
I need to make sure I clearly state that this information is protected under the protective order, but the primary justification for redaction is the potential competitive harm.
**Let me double-check** if there are any exceptions to the protective order that might apply here. I don't think so, but it's always good to be thorough.
```
<end_of_thought>
```
The information that all three parties agree should be redacted from the court's decision is protected information as defined by the protective order. Specifically, the disclosure of information contained in Torres's and G4S's proposals, total evaluated prices, and price-related discussions would provide Torres, G4S, and their competitors with insight into how Torres and G4S approach procurements of local guard services contracts, putting Torres and G4S at a competitive disadvantage. Accordingly, the court will redact that information from its decision.

Figure 2: Example of synthetic data (Law domain). **Black text** represents the original text. **Green text** indicates the hidden thoughts reconstructed from the original text.

**Hidden Thoughts Generation**   We generate hidden thoughts for the preprocessed texts using an LLM. For each sample in both domains (STEM and Law), we add hidden thoughts to the text. Using the prompt in Appendix G, we generate hidden thoughts marked by thought tags (`<start_of_thought>` and `<end_of_thought>`). We use the same prompt for both STEM and Law domains. We then combine the generated hidden thoughts with the original texts to create synthetic data. We use `Gemma2-9B-it` (Shao et al., 2024) for this task, though stronger reasoning models such as Gemini 2.5 Pro (Google, 2025) or DeepSeek-R1 (DeepSeek-AI et al., 2025) can also be used. We set the temperature to 0.3 and limit each hidden thoughts to a maximum of 512 tokens. We repeat sampling when needed to ensure the generated hidden thoughts fit within this limit. Hence, the synthetic sequence—created by concatenating the original text with its hidden thoughts—always fits within the 1 024-token maximum sequence length used during training. The final synthetic texts contained 150,000 examples (85.8M tokens) for STEM and 150,000 examples (66.5M tokens) for Law.

## 2.3   REASONING CPT

Reasoning CPT involves continuing pretraining on text sequences where hidden thoughts come before the original text. From a learning perspective, Reasoning CPT works like standard CPT, using autoregressive language modeling to predict the next token. Formally, it minimizes this loss function:

$$\mathcal{L}_{\text{CPT}}(\theta) = -\mathbb{E}_{X \sim \mathcal{D}} \left[ \sum_{t=1}^{L} \log p_\theta(x_t | x_{<t}) \right] \quad (1)$$

where $\theta$ represents the parameters of the pre-trained model, $\mathcal{D}$ is the training data, $X = (x_1, x_2, \cdots, x_L)$ is a text sequence sampled from this distribution, $x_t$ is the $t$-th token in $X$, and $x_{<t}$ is the token sequence before it. For original text $S = (s_1, s_2, \cdots)$, standard CPT uses training data in the form $X = S$ (training on original text only). Reasoning CPT uses a different format:

$$X = \texttt{<start\_of\_thought>} \oplus H \oplus \texttt{<end\_of\_thought>} \oplus S \tag{2}$$

where $H = (h_1, h_2, \cdots)$ is the token sequence of hidden thoughts without the thought tags, and $\oplus$ means sequence concatenation. Thus, the only difference between Reasoning CPT and CPT is the format of the training data. In our experiments, we did not add `<start_of_thought>` and `<end_of_thought>` as new special tokens to the vocabulary but encoded them using existing subword tokenization. Therefore, the vocabulary size and number of parameters stay the same as in standard CPT.

## 3 EVALUATION OF REASONING CAPABILITIES

In this section, we compare LLMs trained with Reasoning CPT and those trained with standard CPT, evaluating their reasoning capabilities. We analyze the effectiveness of each approach across MMLU and GPQA benchmark.

### 3.1 EXPERIMENTAL SETUP

**Evaluation Targets** Our study aims to examine how including hidden thoughts in continual pretraining affects reasoning performance. We compare the following configurations: (1) base model (`Gemma2-9B`), (2) CPT, and (3) Reasoning CPT. CPT and Reasoning CPT use data built from the same text corpus, with the only difference being whether hidden thoughts is included. We train separate models for STEM and Law domains to also examine how domain characteristics affect reasoning capabilities.

**Training Settings** Due to the high computational cost of training large models, we focus on training and analyzing smaller models. All methods use the same training configuration. We train `Gemma2-9B`[2] (Shao et al., 2024) and `Qwen2.5-7B`[3] (Yang et al., 2024a). For computational efficiency, we fine-tune using LoRA (Hu et al., 2022). All models use the same hyperparameters: LoRA rank $r = 64$, learning rate 3e-5 (with cosine decay schedule), 6 epochs, maximum sequence length 1024, and AdamW (Loshchilov & Hutter, 2019) optimizer. We set batch size 8 for Gemma2-9B and 128 for Qwen2.5-7B. All training was conducted on NVIDIA A100 GPUs.

**Evaluation** For evaluation, we use MMLU (Hendrycks et al., 2021), a comprehensive benchmark covering a wide range of multiple-choice questions across diverse domains, including STEM fields and social sciences such as Law. MMLU is well-suited for evaluating general reasoning ability, as it requires domain-specific knowledge and advanced inference across various disciplines. In addition, we evaluate models on GPQA Diamond (Rein et al., 2023), which is designed to measure graduate-level reasoning in the sciences. GPQA is substantially more challenging than MMLU. GPQA covers three subject areas: Biology, Chemistry, and Physics, and is widely considered a stress test for reasoning ability in STEM domains. For evaluation prompts, we use few-shot examples. To minimize influence from excessive examples, we use 2-shot prompts. The prompts used are detailed in §G.2.

### 3.2 BENCHMARK RESULTS

**Finding 1: Reasoning CPT enables cross-domain reasoning skills** According to the results in Table 1, Reasoning CPT achieved notable performance improvements on MMLU compared to the base models `Gemma2-9B` and `Qwen2.5-7B`. On MMLU, Reasoning CPT trained on STEM domain data showed a 5.4-point improvement in the STEM category (69.4%), along with solid gains in humanities (+2.9 points) and other categories (+2.8 points), achieving an overall improvement of +3.3 points. Similarly, Law-trained Reasoning CPT improved STEM accuracy by 4.3 points

---

[2]`https://huggingface.co/google/gemma-2-9b`
[3]`https://huggingface.co/Qwen/Qwen2.5-7B`

Table 1: Accuracy (%) of each model on MMLU subsets. **Bold** indicates the highest accuracy in each column, underline indicates the second highest. Green and gray values indicate the delta from the base model.

| Method | Domain | STEM | Social Sciences | Humanities | Other | All |
|---|---|---|---|---|---|---|
| Gemma2-9B | - | 64.0 | 74.5 | 57.6 | 71.0 | 65.8 |
| + CPT | Law | 66.2 | 75.9 | 57.8 | 71.5 | 66.7 |
| + CPT | STEM | 67.0 | 74.5 | 59.3 | 72.4 | 67.3 |
| + Reasoning CPT | Law | 68.3 (+4.3) | **77.2** (+2.7) | 59.0 (+1.4) | 72.6 (+1.6) | 68.1 (+2.3) |
| + Reasoning CPT | STEM | **69.4** (+5.4) | 77.1 (+2.6) | **60.5** (+2.9) | **73.8** (+2.8) | **69.1** (+3.3) |
| Qwen2.5-7B | - | 69.9 | 74.2 | 57.3 | 70.8 | 66.8 |
| + CPT | Law | 70.3 | 73.3 | 56.0 | 71.3 | 66.4 |
| + CPT | STEM | 70.4 | 73.4 | 57.0 | 69.8 | 66.5 |
| + Reasoning CPT | Law | **72.7** (+2.8) | **75.2** (+1.0) | **57.9** (+0.6) | **71.6** (+0.8) | **68.0** (+1.2) |
| + Reasoning CPT | STEM | 71.5 (+1.6) | 74.5 (+0.3) | 56.8 (-0.5) | 71.1 (+0.3) | 67.1 (+0.3) |

Table 2: Accuracy (%) of each model on GPQA subsets.

| Method | Domain | Biology | Chemistry | Physics | All |
|---|---|---|---|---|---|
| Gemma2-9B | - | 31.6 | 23.7 | 22.1 | 23.7 |
| + CPT | Law | 26.3 | 24.7 | 29.1 | 26.8 |
| + CPT | STEM | 31.6 | **31.2** | 23.3 | 27.8 |
| + Reasoning CPT | Law | 36.8 (+5.2) | 29.0 (+5.3) | **31.4** (+9.3) | 30.8 (+7.1) |
| + Reasoning CPT | STEM | **52.6** (+21.0) | 31.2 (+7.5) | 30.2 (+8.1) | **32.8** (+9.1) |
| Qwen2.5-7B | - | 26.3 | 20.4 | 30.2 | 25.2 |
| + CPT | Law | 31.6 | 25.8 | 23.3 | 25.3 |
| + CPT | STEM | 31.6 | 24.7 | 24.4 | 25.3 |
| + Reasoning CPT | Law | 26.3 (+0.0) | **31.2** (+10.8) | 30.2 (+0.0) | 30.3 (+5.1) |
| + Reasoning CPT | STEM | **47.4** (+21.1) | 28.0 (+7.6) | **32.6** (+2.4) | **31.8** (+6.6) |

(68.3%), indicating that hidden thought training transfers reasoning skills across domains. According to the results in Table 2, GPQA Diamond is an extremely challenging benchmark, with base models performing at nearly chance level: 23.7% for Gemma2-9B and 25.2% for Qwen2.5-7B. This confirms that GPQA requires substantially more complex reasoning than MMLU. Nevertheless, Reasoning CPT scores were much higher, outperforming both base models and standard CPT by a wide margin. This suggests that hidden thought training provides domain-independent reasoning skills that generalize beyond the original training corpus.

**Finding 2: Reasoning CPT improves model reasoning performance beyond CPT**  Even compared to standard CPT (without hidden thoughts), Reasoning CPT consistently showed better performance across most domains. On MMLU using Gemma2-9B, the STEM-trained CPT achieved 67.3%, while the Reasoning CPT reached 69.1%, marking a 1.8-point improvement. Similarly, the Law-trained CPT scored 66.7%, whereas the Reasoning CPT reached 68.1%, a 1.4-point gain. Notably, the Reasoning CPT trained on legal texts significantly outperformed the STEM-trained CPT. For instance, with Qwen2.5-7B, the Law-trained Reasoning CPT achieved 30.3%, surpassing the STEM-trained CPT baseline of 25.3% by 5.0 points.

One possible concern is that Reasoning CPT introduces additional tokens through hidden thoughts, which could independently contribute to performance gains. To test this, we compared MMLU overall accuracy trajectories with respect to training token count for CPT and Reasoning CPT (Figure 3). In both STEM and Law domains, Reasoning CPT consistently outperformed CPT at the same token count, indicating that the observed gains are attributable to hidden thought training rather than simply increased exposure to tokens.

**Finding 3: Reasoning CPT is especially effective for difficult problems**  Table 3 shows the accuracy of each model on MMLU problems classified into five difficulty levels (Very Easy, Easy,

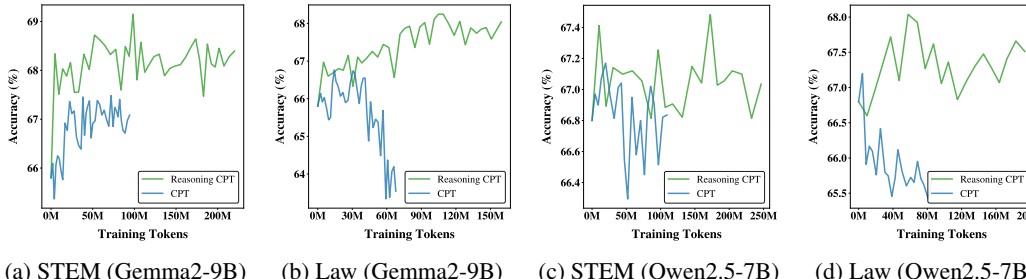

(a) STEM (Gemma2-9B)     (b) Law (Gemma2-9B)     (c) STEM (Qwen2.5-7B)     (d) Law (Qwen2.5-7B)

Figure 3: MMLU accuracy trends with training token count for CPT and Reasoning CPT across two domains (STEM and Law) and two base models. The horizontal axis shows cumulative tokens trained during training (in millions), and the vertical axis shows overall MMLU accuracy (%). Reasoning CPT consistently outperforms CPT, suggesting that accuracy improvements are due to training with hidden thoughts rather than just increased token count.

Table 3: Accuracy (%) for each model across different difficulty levels on MMLU. Green and gray values indicate the delta from the base model (`Gemma2-9B` or `Qwen2.5-7B`).

| Method | Domain | Very Easy | Easy | Medium | Hard | Very Hard |
|---|---|---|---|---|---|---|
| `Gemma2-9B` | - | 83.9 | 73.1 | 67.0 | 52.0 | 41.3 |
| + CPT | Law | 82.4 | 72.8 | 66.1 | 53.3 | 44.6 |
| + CPT | STEM | 83.8 | 73.8 | 66.9 | 52.7 | 43.9 |
| + Reasoning CPT | Law | 82.9 (−1.0) | 73.0 (−0.1) | 68.1 (+1.1) | 56.0 (+4.0) | 52.5 (+11.2) |
| + Reasoning CPT | STEM | 85.1 (+1.2) | 73.7 (+0.6) | 69.6 (+2.6) | 56.8 (+4.8) | 51.8 (+10.5) |
| `Qwen2.5-7B` | - | 80.0 | 73.5 | 67.5 | 54.7 | 46.8 |
| + CPT | Law | 82.6 | 72.9 | 66.6 | 54.7 | 42.5 |
| + CPT | STEM | 79.4 | 72.9 | 67.2 | 54.1 | 43.9 |
| + Reasoning CPT | Law | 81.8 (+1.8) | 73.0 (−0.5) | 67.9 (+0.4) | 54.8 (+0.1) | 51.1 (+4.3) |
| + Reasoning CPT | STEM | 80.6 (+0.6) | 72.5 (−0.9) | 67.5 (+0.0) | 54.8 (+0.1) | 49.6 (+2.9) |

Medium, Hard, Very Hard) by GPT-4o. The prompt used for difficulty classification is in §G.3. These results show that Reasoning CPT's effect becomes more pronounced as difficulty increases. For Very Easy problems, there is only a marginal difference between the base model `Gemma2-9B` and each method. However, as difficulty increases, the advantage of Reasoning CPT grows. Particularly notable is the performance on Very Hard problems. In the STEM domain, Reasoning CPT achieved 51.8% compared to the base model's 41.3%, a substantial 10.5-point improvement. Similarly, in the Law domain, performance improved from 41.3% to 52.5%, an 11.2-point gain. Comparing standard CPT with Reasoning CPT, the difference in the STEM domain is 7.9 points (43.9% → 51.8%), and in the Law domain also 7.9 points (44.6% → 52.5%), showing about an 8-point improvement in both domains. These results suggest that explicitly learning hidden thoughts is particularly effective for problems requiring sophisticated reasoning.

## 4 ANALYSIS OF SYNTHETIC DATA WITH HIDDEN THOUGHTS

The training data for Reasoning CPT consists of synthetic samples that pair original text with generated hidden thoughts. The length of hidden thoughts produced during inference is a critical factor for model performance, and it is strongly influenced by the distribution of hidden thought lengths in the training data. In this section, we analyze how much the length of the original text explains the length of hidden thoughts, and how this relationship differs across domains, specifically STEM and Law.

**Setup** For 150,000 synthetic samples from each domain, we computed the Spearman rank correlation coefficient between the number of tokens in the original text and the number of tokens in the generated hidden thoughts. A positive correlation indicates that longer texts tend to yield longer

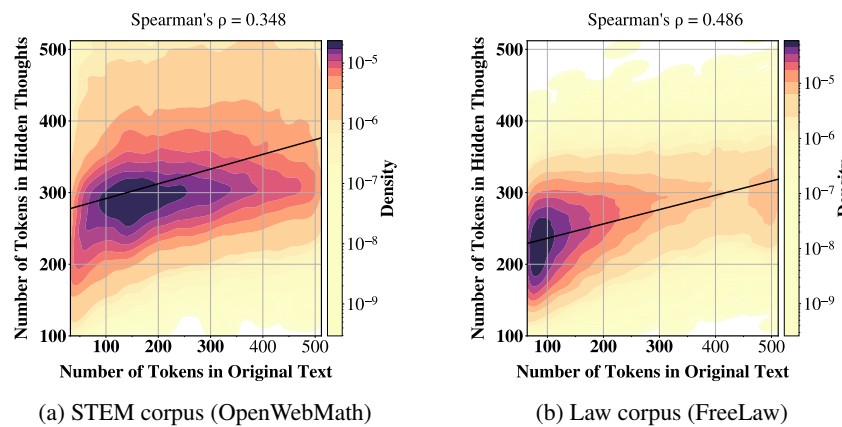

(a) STEM corpus (OpenWebMath)  (b) Law corpus (FreeLaw)

Figure 4: Correlation between original text token count and hidden thoughts token count

Table 4: Average and standard deviation of hidden thought token counts, grouped by the length of the original text (the corpus given to the LLM to generate hidden thoughts) in tokens, for STEM and Law. Values are presented as mean $\pm$ std.

| Original text length range (tokens) | Law | STEM |
|---|---|---|
| 0–100 | $224.86 \pm 44.90$ | $273.46 \pm 80.20$ |
| 100–200 | $248.36 \pm 45.56$ | $307.14 \pm 72.00$ |
| 200–300 | $272.66 \pm 44.73$ | $326.82 \pm 72.56$ |
| 300–400 | $289.46 \pm 44.85$ | $343.97 \pm 80.10$ |
| 400–500 | $299.09 \pm 44.42$ | $356.60 \pm 87.04$ |
| 500–600 | $296.06 \pm 44.37$ | $362.26 \pm 93.05$ |

hidden thoughts. In addition, we divided the original text length into ranges of 100 tokens and measured the mean and standard deviation of hidden thought length within each range.

If thought lengths vary widely within the same text length range (high standard deviation), this implies that factors beyond text length, such as content complexity, play an important role. Conversely, a small variance suggests that text length alone explains thought length reasonably well.

**Finding 1: Longer texts yield longer hidden thoughts across domains**  Figure 4 shows the relationship between the number of tokens in the original text and the number of tokens in the generated hidden thoughts. Both domains exhibit positive correlations, with Spearman's $\rho = 0.348$ for STEM and $\rho = 0.486$ for Law. These results clearly demonstrate that longer texts tend to generate longer hidden thoughts. At present, the dominant method for obtaining long reasoning trajectories is to collect derivations from solving mathematics problems. However, the presence of long texts in domains such as Law indicates that long hidden thoughts can also be generated in other fields. This suggests that long reasoning trajectories can be constructed from diverse domains, rather than relying solely on mathematics.

**Finding 2: STEM texts show greater variance in thought length than Law texts**  Table 4 reports the mean and standard deviation of hidden thought token counts across text length ranges. In the Law domain, the variance is highly stable (std $\approx$ 44–45), showing a consistent relationship between text length and thought length. In contrast, the STEM domain shows large fluctuations, with especially high variability for short texts (0–100 tokens, std = 80.20). These differences reflect the characteristics of each domain. In STEM texts, particularly mathematics and science, concise formulas or theorems often conceal lengthy derivation processes. As a result, short inputs can trigger disproportionately long hidden thoughts, leading to higher variance. For example, as shown in Figure 12, even short mathematical expressions produce very long hidden thoughts. In contrast, Law texts tend to align more directly with the complexity of their reasoning, resulting in stronger correlation coefficients and more stable variance.

## 5 RELATED WORK

**Post-training methods**  Research on enhancing the reasoning capabilities of LLMs has been developing rapidly in recent years. Approaches to strengthen reasoning ability include repeated sampling (Wang et al., 2023; Cobbe et al., 2021; Brown et al., 2024), self-correction (Madaan et al., 2023; Shinn et al., 2023; Chen et al., 2024b), and tree search (Yao et al., 2023; Qi et al., 2024), but we focus on post-training methods. The most common approach for enhancing reasoning capabilities in post-training is SFT of pre-trained models. For example, STaR (Zelikman et al., 2022) generates and trains with CoT. s1 (Muennighoff et al., 2025) performs SFT with about 1,000 carefully selected examples. Additionally, RL approaches, as represented by o1 (Jaech et al., 2024) and r1 (DeepSeek-AI et al., 2025), have significantly contributed to reasoning capability improvement. Continual pretraining is also known to improve LLM reasoning capabilities. Many previous studies have shown that pretraining with STEM corpora is effective (Yang et al., 2024b; Aryabumi et al., 2024; Petty et al., 2024; Kim et al., 2024; Uchiyama et al., 2024). In the Law domain, Zhang et al. (2025) improved LLM's legal mathematical capabilities using semi-automatically constructed legal data.

**Enhancing reasoning capability during continual pretraining**  Attempts to explicitly train hidden thoughts from the continual pretraining stage are still limited, but notable research has recently emerged. For example, Quiet-STaR (Zelikman et al., 2024) presents a new framework for acquiring thinking tokens through RL. Their algorithm marks an important step toward acquiring thinking generation capability itself, but we focus on training data construction, making our approaches complementary. Recently, Ruan et al. (2025) show that reasoning capabilities can be significantly improved by learning synthetic data that includes potential latent thoughts behind text in the mathematics domain. Building on these insights, our study is the first systematic analysis of the effects of continual pretraining incorporating hidden thoughts across different domains: STEM and Law. We particularly emphasize modeling realistic reasoning processes, including expert background knowledge recall, self-verification, and human-like thinking styles. By comparing and analyzing standard CPT and Reasoning CPT, we provide new insights into how continual pretraining with hidden thoughts affects reasoning ability, examining domain differences and thinking efficiency aspects.

**Synthetic data**  Rajani et al. (2019) shows that training LLMs with text containing human-created annotations improves common sense reasoning performance. For synthetic data generation from text, useful approaches have been proposed, such as automatic generation of instruction data (Chen et al., 2024a) and frameworks for inserting thinking within context (Lanchantin et al., 2023). Our study shares the basic approach of deriving useful information from high-quality text and applying it to learning. However, we focus on continual pretraining with explicitly added hidden thoughts.

## 6 CONCLUSION

This study explored how adding hidden thoughts to continual pretraining affects LLM performance. We found several important results. These findings suggest three promising future directions. First, post-training Reasoning CPT models with RL or SFT could lead to even better performance. Reasoning CPT builds strong reasoning foundations that RL and SFT could further refine. Second, while we focused on STEM and Law domains, this approach could be applied to many other fields where designing strict reward signals is difficult. For example, by learning the hidden thoughts behind novels, models may improve their creativity and narrative reasoning in creative writing domains such as fiction. Similarly, learning the hidden thoughts behind scientific papers may enhance their ability to explore scientific ideas. Third, while we generated hidden thoughts for human-written texts, Reasoning CPT could also be applied to synthetic data created by LLMs. As continual pretraining increasingly shifts toward synthetic data, adding hidden thoughts could significantly improve model performance.

## REPRODUCIBILITY STATEMENT

Details necessary for reproducibility are provided throughout the paper. The procedure for generating synthetic data is in § 2 and Appendix G. Training settings, hyperparameters, information about computational resources are described in §3.1. Evaluation prompts are included in §G.2.

## ETHICS STATEMENT

This study does not involve human subjects, personal data, or sensitive information. All training corpora are publicly available datasets, and hidden thoughts were synthetically generated using large language models. However, synthetic data may inherit biases specific to the underlying domains. In particular, because the data generation method described in this paper is not limited to any specific domain, it could also be applied to texts that contain political, cultural, or gender-related biases, thereby creating biased synthetic training data for LLMs. Moreover, even when such biased texts are not used, the latent biases inherent in LLMs themselves may still be reflected in the generated synthetic data.

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

## A    ANALYSIS OF REASONING EFFICIENCY

Longer chains of thought improve accuracy but incur greater computational cost and latency (Sui et al., 2025). The "overthinking phenomenon"—spending too many tokens on simple problems—is a practical challenge for reasoning models. Efficient reasoning, balancing thinking token count with reasoning accuracy, is a key design element in LLM reasoning. If reasoning is too verbose, it wastes computational resources; if too brief, accuracy suffers. In practical scenarios, models must handle problems of varying difficulty, making appropriate adjustment of reasoning depth essential. This section analyzes how much thought each model generates for problems of different difficulty levels, analyzing the relationship between reasoning efficiency and performance. We focus on these questions: (1) How does reasoning length change with problem difficulty? (2) What creates the difference in reasoning efficiency between standard CPT and Reasoning CPT? To answer these questions, we analyze the relationship between thinking token count and accuracy using the MMLU problems and model outputs from §3.2.

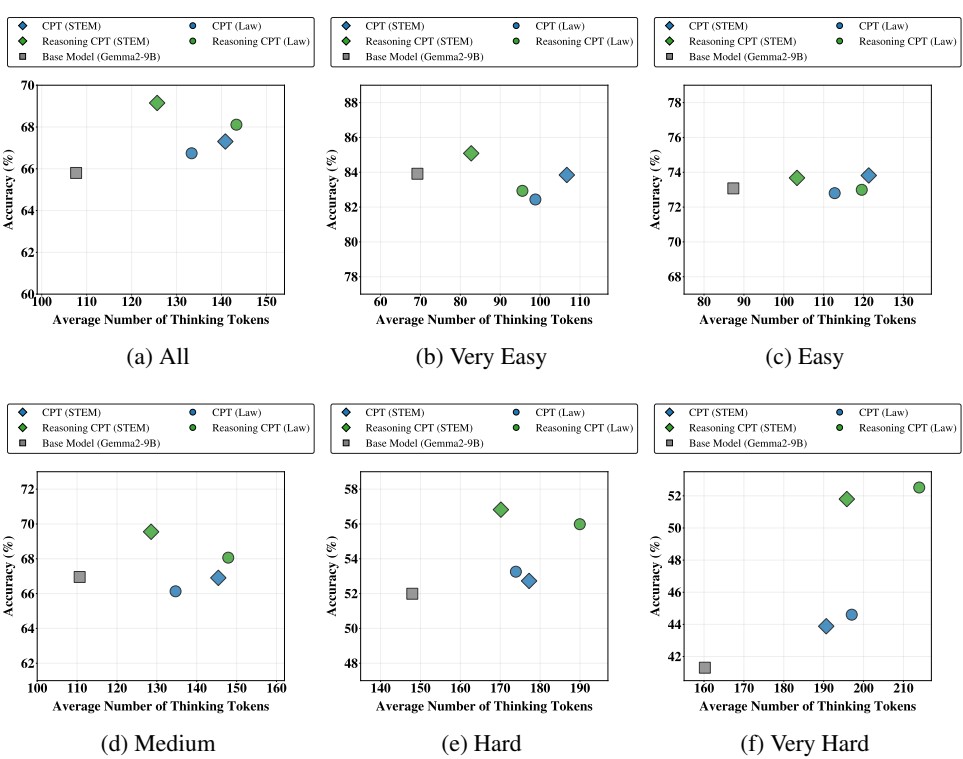

Figure 5: Relationship between problem difficulty, length of generated hidden thoughts, and accuracy. Reasoning CPT models dynamically adjust the length of their generated thoughts based on problem difficulty: they use fewer tokens than standard CPT on easy problems, and use more tokens on hard problems with substantial accuracy gains.

### A.1    RESULTS

To examine how Reasoning CPT affects reasoning efficiency during inference, we analyze how the token count of hidden thoughts (enclosed by `<start_of_thought>` and `<end_of_thought>` tags) changes in inference. Figure 5 compares the token count and accuracy at each difficulty level of MMLU across five models based on `Gemma2-9B`. This figure shows that models trained with Reasoning CPT appropriately adjust the length of their generated thoughts based on problem difficulty. For simpler problems (Very Easy, Easy), Reasoning CPT models achieve higher accuracy with fewer thinking tokens, while for more difficult problems (Hard, Very Hard), they significantly increase thought length with corresponding substantial accuracy improvements. At **Very Easy** and **Easy** difficulty levels, Reasoning CPT uses fewer thinking tokens than CPT in both domains, with little difference in accuracy. At **Medium** difficulty, Reasoning CPT models substantially increase token

count while CPT models show relatively smaller increases. Accordingly, the accuracy gap between Reasoning CPT and CPT begins to widen, with Reasoning CPT scoring 2.6 points higher in STEM and 1.1 points higher in Law. The differences between models become most pronounced at **Hard** and **Very Hard** difficulty levels. From Hard to Very Hard problems, Reasoning CPT models tend to increase thought length, with corresponding substantial accuracy improvements. For example, at Very Hard difficulty, Reasoning CPT-trained models achieve 51.8% to 52.5% accuracy, while CPT accuracy remains at 43.9% to 44.6%. This approximately 8-point accuracy difference suggests that models trained with Reasoning CPT have acquired the ability to automatically think longer for complex problems.

A plausible driver of this behaviour is the positive correlation between the length of the original text and the length of its hidden-thought segment observed in our training corpus.

## B    ANALYSIS OF REASONING DIVERSITY

This section analyzes how thought diversity generated by models contributes to reasoning performance. We focus on whether sampling increases the number of solvable problems, evaluated through the improvement in Pass@$k$ (Chen et al., 2021). Pass@$k$ is a metric that represents the probability of a model getting at least one correct answer when attempting the same problem $k$ times. This metric is an important measure of a model's ability to generate diverse solutions to problems. If the Pass@$k$ value increases significantly as $k$ increases, it indicates that the model has diverse reasoning paths and can reach correct answers through different reasoning attempts.

### B.1    SETUP

In this section, we use GSM8k (Cobbe et al., 2021), a mathematical reasoning task that requires generating numerical answers, to properly evaluate thought diversity. This is because multiple-choice tasks like MMLU would naturally increase the probability of including correct answers as sampling increases, making it unsuitable as a metric for evaluating true thought diversity. The models evaluated include `Gemma2-9B`, its instruction-tuned version `Gemma2-9B-it`, and Reasoning CPT. We sample with a temperature of 0.3 and observe the changes in Pass@$k$. We use zero-shot for `Gemma2-9B-it`, and 1-shot for `Gemma2-9B` and Reasoning CPT. Details of the 1-shot prompt are shown in §G.4.

### B.2    RESULTS

The instruction-tuned model (`Gemma2-9B-it`) shows high initial accuracy at Pass@1 (81.2%), with an improvement of merely 3.8 points to Pass@5. This indicates that this model strongly converges on specific reasoning paths and can hardly find new reasoning paths even with increased sampling. This trend is also demonstrated in Yue et al. (2025), which re-examines the effects of reinforcement learning with verifiable reward (RLVR) (Lambert et al., 2024; DeepSeek-AI et al., 2025), showing that many outputs obtained through RLVR already exist in the base model, and RLVR merely redistributes the probability distribution of existing outputs rather than introducing new reasoning paths. Similarly, Tang et al. (2025) shows that models optimized for inference-time objectives like Pass@$k$ without careful consideration tend to exploit existing outputs rather than generate fundamentally new reasoning paths. These findings support the idea that instruction tuning tends to narrow the output distribution.

In contrast, the base model (`Gemma2-9B`) shows large improvements from Pass@1 to Pass@5, and the Reasoning CPT model similarly shows large Pass@$k$ scaling. This suggests that these models maintain diversity in their output distributions. Importantly, Reasoning CPT surpasses `Gemma2-9B-it`'s Pass@5 at just Pass@2, and ultimately achieves an accuracy of 91.7% at Pass@5. These results indicate that to solve more problems, it is essential for the base model itself to have the ability to generate diverse reasoning paths. Reasoning CPT not only preserves such diversity but even enhances it to a higher level compared to the base model, making it an effective approach for improving reasoning capabilities through diverse thought generation.

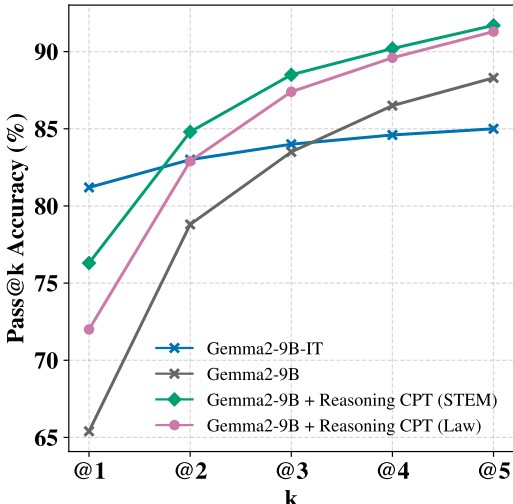

Figure 6: Comparison of Pass@$k$ accuracy for each model on GSM8k. `Gemma2-9B` and Reasoning CPT show large improvements as $k$ increases, indicating more problems can be solved with more sampling attempts. In contrast, the instruction-tuned model (`Gemma2-9B-it`) shows high initial Pass@1 accuracy but small improvements with increased $k$, with accuracy converging at $k = 4$ or $k = 5$. This indicates that increasing sampling attempts hardly adds new reasoning paths.

## C    ANALYSIS OF THOUGHT STYLES

This section analyzes how different thought formats (hidden thoughts style vs. standard CoT) affect reasoning accuracy. We focus on how hidden thoughts style changes reasoning performance compared to standard CoT.

**Setup**    We used `Gemma2-9B` and Reasoning CPT as the evaluation models, testing them on the GSM8k dataset. We applied standard CoT style (5-shot) and hidden thoughts style (1-shot) to each model for comparison. Details of the prompts are shown in § G.5 and § G.4. Following the setup in (Ruan et al., 2025), we excluded the thought tags (`<start_of_thought>` and `<end_of_thought>`) in the 5-shot CoT prompts for `Gemma2-9B`. We sampled with a temperature of 0.3.

Table 5: GSM8k Accuracy Comparison by Thought Style (%)

| Model | 5-shot CoT | 1-shot Hidden Thoughts |
|---|---|---|
| `Gemma2-9B` | 58.3 | 65.4 |
| + Reasoning CPT (STEM) | 69.5 | 76.3 |
| + Reasoning CPT (Law) | 64.3 | 72.0 |

**Results**    Table 5 shows the accuracy for each model by thought style. hidden thoughts style (1-shot) achieves higher accuracy than standard CoT (5-shot) in all evaluated models. Specifically, with `Gemma2-9B`, hidden thoughts style (1-shot) showed 7.1 percentage points higher accuracy (65.4% vs. 58.3%) than standard CoT (5-shot). Furthermore, when applying hidden thoughts style to Reasoning CPT models, they achieved high accuracy of 76.3% in the STEM domain and 72.0% in the Law domain. This suggests that the combination of hidden thoughts format and continual pretraining significantly improves reasoning accuracy. An important observation is that hidden thoughts style shows higher performance with fewer shots (1-shot vs. 5-shot) than standard CoT, indicating that the hidden thoughts format contributes to the acquisition of efficient thought processes.

## D  LOSS TRENDS DURING CONTINUAL PRETRAINING

We compare Reasoning CPT and standard CPT under the same continual pretraining settings. Figure 7 shows the training loss observed during this process. As illustrated, Reasoning CPT consistently achieves lower loss than standard CPT, both in terms of training steps and the number of training tokens.

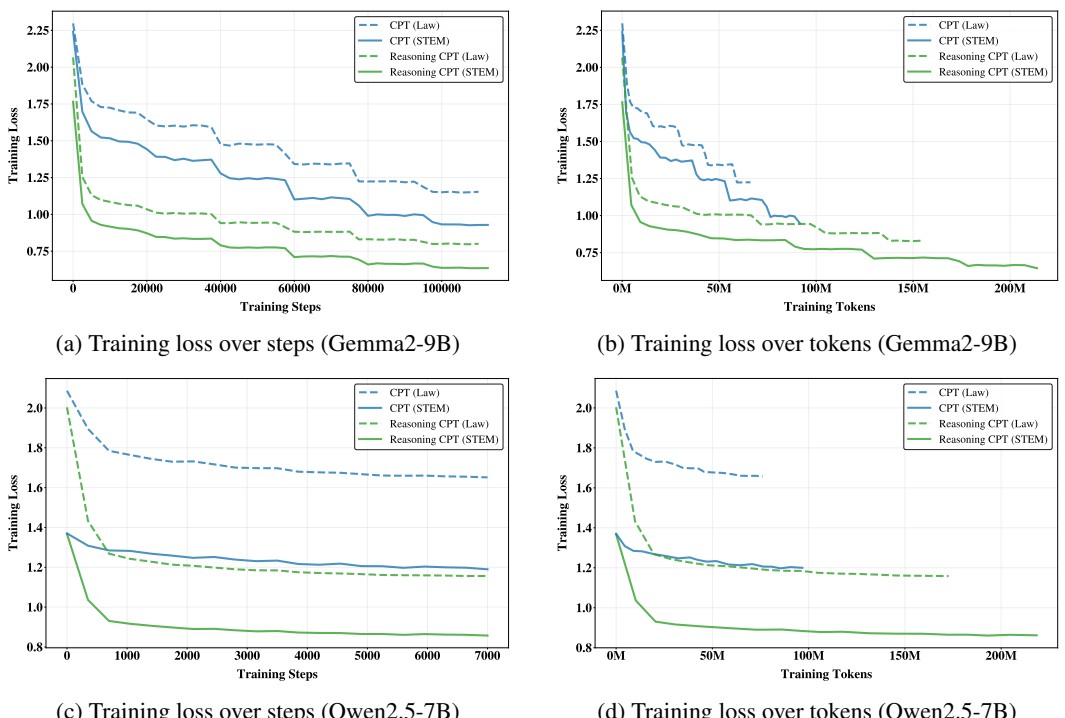

(a) Training loss over steps (Gemma2-9B)  (b) Training loss over tokens (Gemma2-9B)

(c) Training loss over steps (Qwen2.5-7B)  (d) Training loss over tokens (Qwen2.5-7B)

Figure 7: Training loss observed during continual pretraining on Gemma2-9B and Qwen2.5-7B. Left plots show the loss over training steps, and right plots show the loss over the number of training tokens. In both models, Reasoning CPT consistently shows lower loss than standard CPT, suggesting that hidden thoughts accelerate convergence.

## E  DOMAIN-SPECIFIC REASONING EFFICIENCY

Table 6 compares the average number of thinking tokens generated during training and inference.

First, consider the domain differences in the *training* data. Hidden thoughts sampled from STEM texts contain more tokens on average (320.7) than those from Law texts (254.1). During *inference* on MMLU, however, the pattern reverses: the STEM-trained model produces shorter thoughts (125.7) than the Law-trained model (143.3).

Several factors may account for this reversal. One possibility is that it reflects fundamental characteristics of the domains themselves. As shown in Table 6, sampled STEM thoughts have both a higher mean and a larger variance. Mathematics and physics literature often compress many implicit reasoning steps into concise theorems; exposing the model to such diversity may have taught it to reason more efficiently. Another possibility is that Reasoning CPT on STEM data encourages the model to discover domain-specific shortcuts, or that the broader token-count diversity of the original STEM texts exerts a regularising effect.

Table 6: Comparison of average thinking token count during inference and training (mean $\pm$ standard deviation). The STEM domain tends to produce shorter thinking tokens during inference compared to the Law domain, despite having longer thinking tokens in the training data.

| | Thinking Tokens (mean±std) | | |
| --- | --- | --- | --- |
| | STEM model | | Law model |
| Training (synthetic data) | $320.7 \pm 81.8$ | > | $254.1 \pm 51.3$ |
| Inference (MMLU) | $125.7 \pm 61.5$ | < | $143.3 \pm 59.7$ |

## F EXAMPLE OF SYNTHETIC DATA WITH HIDDEN THOUGHTS

<start_of_thought>
Right, I need to explain why we don't see fractional charges directly, even though quarks have them.
Hmm, the key is that quarks are fermions, and fermions obey the Pauli exclusion principle. This means their wavefunction must be antisymmetric under particle exchange. But if quarks were the only fundamental particles, we'd see fractional charges directly.
Wait, I remember! There's a solution to this puzzle: color charge.
Let me start by mentioning the experimental evidence for quarks—deep inelastic scattering reveals these "pips" inside nucleons, which are identified as quarks. But we don't see individual quarks directly.
Then, I should explain the issue with the wavefunction. If quarks are fermions with spin 1/2, their wavefunction needs to be antisymmetric. But if they all have the same flavor and spin, their spatial wavefunction would be symmetric, leading to a total symmetric wavefunction. That's a problem!
Aha, the solution is to introduce color charge. We assume quarks have an additional quantum number called "color," which comes in three types. This allows us to make the total wavefunction antisymmetric by including a color wavefunction that's antisymmetric under quark exchange.
This introduces a new symmetry, SU(3), which is related to the gauge symmetry of the strong interaction, QCD.
So, we've shifted the question from "why don't we see fractional charges?" to "why can't we see colored particles?"
That's a good place to leave it for now.
<end_of_thought>
So why don't we see fractional charges in nature? This is an important point! In so-called deep inelastic scattering we see pips inside the nucleon – these have been identified as the quarks. We do not see any direct signature of individual quarks. Furthermore, if quarks are fermions, as they are spin $1/2$ particles, what about antisymmetry of their wavefunction? Let us investigate the $\Delta^{++}$, see Fig. [fig:D++], which consists of three $u$ quarks with identical spin and flavour (isospin) and symmetric spatial wavefunction, $\psi_{\text{total}} = \psi_{\text{space}} \times \psi_{\text{spin}} \times \psi_{\text{flavour}}$. This would be symmetric under interchange, which is unacceptable. Actually there is a simple solution. We "just" assume that there is an additional quantity called colour, and take the colour wave function to be antisymmetric: $\psi_{\text{total}} = \psi_{\text{space}} \times \psi_{\text{spin}} \times \psi_{\text{flavour}} \times \psi_{\text{colour}}$ We assume that quarks come in three colours. This naturally leads to yet another $SU(3)$ symmetry, which is actually related to the gauge symmetry of strong interactions, QCD. So we have shifted the question to: why can't we see coloured particles?

Figure 8: Example of synthetic data (STEM domain) with hidden thoughts

<start_of_thought>
OK, let's figure out where the function $f(x) = x^3 - 7x^2 + 15x - 2$ is increasing.

Hmm, remember that a function is increasing when its slope is positive. And the slope is given by the derivative. So, first, I need to find the derivative of $f(x)$.

Right, the power rule tells me that the derivative of $x^n$ is $nx^{n-1}$. Applying that to each term, I get:

$$f'(x) = 3x^2 - 14x + 15$$

Now, I need to find where $f'(x) > 0$. This means I need to solve the inequality:

$$3x^2 - 14x + 15 > 0$$

Hmm, this looks like a quadratic inequality. I could try to find the roots of the quadratic equation $3x^2 - 14x + 15 = 0$ and then analyze the sign of the expression in the intervals determined by the roots.

Let me think... I could use the quadratic formula to find the roots:

$$x = \frac{-b \pm \sqrt{b^2 - 4ac}}{2a}$$

where $a = 3$, $b = -14$, and $c = 15$.

Wait, let me double-check the discriminant, $b^2 - 4ac$.

$$(-14)^2 - 4(3)(15) = 196 - 180 = 16$$

Since the discriminant is positive, there are two real roots. That means the parabola represented by $3x^2 - 14x + 15$ intersects the x-axis at two points.

OK, now I can use the roots to divide the number line into intervals and test the sign of $3x^2 - 14x + 15$ in each interval.

Then, the intervals where $3x^2 - 14x + 15 > 0$ will correspond to the intervals where $f(x)$ is increasing.

The prompt also asks to check the answer with a graph. That's a good idea to visually confirm the analytical result.
<end_of_thought>
Home > APCALC > Chapter 4 > Lesson 4.3.1 > Problem4-113

4-113.

Without graphing, analytically determine where the function $f(x) = x^3 ˇ 7x^2 + 15x ˇ 2$ is increasing. Check your answer with a graph.

If a function is increasing, then its slopes are positive.

FInd where the derivative is greater than 0.

Figure 9: Example of synthetic data (STEM domain) with hidden thoughts

<start_of_thought>
OK, let's write a paper about improving patient queue management. Hmm, what are the main problems with existing systems? Right, they usually don't account for patients who miss their appointments. That can lead to wasted time and frustration for both patients and staff. I need to review existing literature to see what approaches have been tried before. Wait, there must be some research on queue management in healthcare settings. I should look for papers on hospital waiting times, appointment scheduling, and patient flow. Aha, I found some papers discussing different queueing models, but they mostly focus on optimizing service times and resource allocation. None of them seem to address the issue of absent patients directly. That's interesting. It seems like there's a gap in the literature. Maybe I can focus on developing a new model that specifically handles absent patients. I need to think about how to incorporate the concept of "slot-back" into a queueing model. What data would be helpful? I should look at the University of Maiduguri Medical Centre's records. I need to know things like the average consultation time, the number of doctors on duty, and the frequency of patient absences. Right, I can collect data on the longest average time of absence (LATA), average consultation time (ACT), and number of doctors on duty (NDD). If I can figure out how these factors relate to each other, I might be able to develop a formula that predicts the impact of patient absences on queue length and waiting times. Hmm, let me see... maybe I can use a simple linear equation to start with.

$$SbP = LATA(1/(\frac{ACT}{NDD}) + c$$

That looks promising. I need to test this equation with the collected data and see how well it predicts real-world queue behavior. If it works, I can publish a paper describing the new "slot-back" model and its potential benefits for patient queue management.
<end_of_thought>
Enhanced Patient Queue Management: Development Of Slot-Back Model Equation Using University Of Maiduguri Medical Centre As Experimental Site
Aboaba A. Abdulfattah, Abideen A. Ismail, P. Y. Dibal Published 2017 · Medicine
The review of existing queue management systems shows that the systems, though vary in concept, sophistication, and area of application, but all without exception lack the ability to consider an absent queuee. Therefore, the work presented in this paper is basically designed to handle cases of absence on queue due to pressing needs. This is to be achieved by slotting back absentees into the queue using our model equation. The research work reported here uses experimental data obtain through field work and survey at the University of Maiduguri medical centre to arrive at terms such as longest average time of absence (LATA), average consultation time (ACT) and number of doctors on duty (NDD) resulting into a Slot-back model equation thus:

$$SbP = LATA(1/(\frac{ACT}{NDD}) + c$$

. This paper references
Some data provided by SemanticScholar

Figure 10: Example of synthetic data (STEM domain) with hidden thoughts

<start_of_thought>
OK, let's prove Lemma 73.7.1.
First, I need to understand the setup. We have an fpqc covering of algebraic spaces $\{f_i : T_i \to T\}$ and open subspaces $W_i \subset T_i$ that satisfy a certain condition. The goal is to show that there's a unique open subspace $W \subset T$ such that $W_i = f_i^{-1}(W)$ for all $i$.
Hmm, what's the best way to approach this? I remember that fpqc coverings are related to schemes, so I should probably start there.
Aha! Lemma 72.9.5 in Topologies on Spaces tells us we can assume each $T_i$ is a scheme. That simplifies things. Now, let's introduce a scheme $U$ and a surjective étale morphism $U \to T$. This gives us a new fpqc covering $\{T_i \times_T U \to U\}$ where each $T_i \times_T U$ is also a scheme.
Now, the collection of opens $W_i \times_T U$ should correspond to a unique open subscheme $W' \subset U$ by Descent, Lemma 35.13.6. This is where things start to look promising.
Since $U \to T$ is open, we can define $W \subset T$ as the image of $W'$. This seems like it could work!
Wait, I need to verify that $W_i = f_i^{-1}(W)$ for each $i$. That's probably a straightforward check using the properties of pullbacks and open subspaces.
OK, I think I have the main steps. I'll write them out formally and fill in the details.
<end_of_thought>
Lemma 73.7.1. Let $S$ be a scheme. Let $\{f_i : T_i \to T\}_{i \in I}$ be an fpqc covering of algebraic spaces over $S$. Suppose that for each $i$ we have an open subspace $W_i \subset T_i$ such that for all $i, j \in I$ we have $\mathrm{pr}_0^{-1}(W_i) = \mathrm{pr}_1^{-1}(W_j)$ as open subspaces of $T_i \times_T T_j$. Then there exists a unique open subspace $W \subset T$ such that $W_i = f_i^{-1}(W)$ for each $i$.
Proof. By Topologies on Spaces, Lemma 72.9.5 we may assume each $T_i$ is a scheme. Choose a scheme $U$ and a surjective étale morphism $U \to T$. Then $\{T_i \times_T U \to U\}$ is an fpqc covering of $U$ and $T_i \times_T U$ is a scheme for each $i$. Hence we see that the collection of opens $W_i \times_T U$ comes from a unique open subscheme $W' \subset U$ by Descent, Lemma 35.13.6. As $U \to X$ is open we can define $W \subset X$ the Zariski open which is the image of $W'$, see Properties of Spaces, Section 65.4. We omit the verification that this works, i.e., that $W_i$ is the inverse image of $W$ for each $i$. $\square$

Figure 11: Example of synthetic data (STEM domain) with hidden thoughts

<start_of_thought>
OK, let's explain the relationship between $O(n \log(\sqrt{n}))$ and $O(n \log(n))$.
Right, Big O notation is all about how the runtime or space complexity of an algorithm scales as the input size grows. It focuses on the dominant terms and ignores constant factors.
So, I need to compare the growth rates of these two functions as $n$ gets really large. I could try to analyze them directly, but there's a more elegant way.
Wait, I remember! We can use limits to show that they belong to the same class. If the limit of the ratio of the two functions as $n$ approaches infinity is a constant, they have the same asymptotic behavior.
Let's see...

$$\lim_{x \to \infty} (n \log(\sqrt{n}))/(n \log(n))$$

I can simplify this:

$$\lim_{x \to \infty} (n \log(\sqrt{n})/(n \log(n)) = \lim_{x \to \infty} 1/2 * (n \log(n))/(n \log(n)) = 1/2$$

Aha! The limit is a constant, 1/2. This means that $O(n \log(\sqrt{n}))$ and $O(n \log(n))$ are equivalent in terms of Big O notation. They grow at the same rate, just with a constant factor difference.
That makes sense intuitively too—$n \log(\sqrt{n})$ is just a slightly slower growing version of $n \log(n)$.
OK, I think that explains it clearly.
<end_of_thought>
(diff) ← Older revision | Latest revision (diff) | Newer revision → (diff)
$O(n \log(\sqrt{n}))$ and $O(n \log(n))$ belongs to same class of function with respect to Big O notation. There is no difference between them other than a constant factor. $\lim_{x \to \infty} (n \log(\sqrt{n}))/(n \log(n))$ $= \lim_{x \to \infty} 1/2 * (n \log(n))/(n \log(n)) = 1/2$

Figure 12: Example of synthetic data (STEM domain) with hidden thoughts

<start_of_thought>
OK, I need to write the court's opinion denying the petition for rehearing.
First, I need to understand the arguments made by the appellees. They claim any error in the cross-examination of Briggs is harmless because another witness testified about his duties.
Hmm, but the court seems to disagree. They point out that Briggs admitted to driving the car in his testimony, which was potentially prejudicial.
Right, I remember the rules of evidence. Cross-examination can be tricky, especially when dealing with depositions.
Wait, what are the rules regarding offering a deposition during cross-examination? I need to make sure I'm accurate.
Aha! The court is saying that even if the cross-examiner didn't intend to elicit those answers, they still benefited from the improper cross-examination.
This is important because the court is setting a precedent here. They're saying that even if a party accidentally violates the rules of evidence, they can't just offer the deposition later to "cure" the error.
The court's reasoning is sound. It emphasizes the importance of following the rules of evidence during cross-examination.
I need to make sure my writing clearly explains the court's reasoning and the specific arguments made in the petition.
<end_of_thought>
PER CURIAM. Appellees vigorously insist, in their petition for rehearing, that any error in the cross-examination of Briggs is harmless, because another witness testified as to his duties, without contradiction. If that had been all to which Briggs testified, the point would be well taken. A reading of the record left the impression with a majority of this court that Briggs admitted that he was driving the car at the time of the accident. One such statement of Briggs was pointed out in the opinion; there are other similar statements in the record. If members of this court gained that impression from a reading of the record, members of the jury doubtless gathered the same impression from a reading of the deposition. Counsel assures us that he did not intentionally elicit these answers from the witness, as he desired to stay away from that pivotal point. We accept his statement. The fact remains that the answers were made in the course of a cross-examination that was in direct violation of the rules of evidence. Intentionally or not, counsel availed himself of the benefit of such testimony. We have no quarrel with the contention, ably presented, that the cross-examiner may offer the whole of a deposition, or the cross-examination, if the proponent of the witness does not. We differ with his contention that a cross-examination, improper at the time, and inadmissible if the proponent offers the deposition, is proper and admissible if, perchance, the cross-examiner offers it first. If this contention is sound, then a cross-examiner need pay no heed to the rules of evidence when a deposition is taken, for any transgression may be cured by the simple device of offering the deposition as his own. The petition for rehearing is denied. COTTERAL, Circuit Judge, dissents.

Figure 13: Example of synthetic data (Law domain) with hidden thoughts

```
<start_of_thought>
```
OK, this is a patent case about whether an appeal can be taken from a Patent Examiner's "withdrawal" of claims.

First, I need to understand the procedural history. The Examiner rejected some claims and "withdrew" others, which meant they were not considered for now but could be refiled. The plaintiff tried to appeal this "withdrawal" but the District Court dismissed it, saying it wasn't a final decision.

Wait, what's this "withdrawal" thing? I need to look up the practice and its legal implications.

Ah, I remember reading about it before. It's a somewhat informal procedure in the Patent Office, but its legal status has been unclear.

Aha! *In re Haas* is relevant here. It held that a "withdrawal" is actually a rejection if it's based on independent inventions. That means it's appealable!

But how does *Hengehold* fit in? It seems to set a standard for what constitutes an "adverse decision" reviewable by the board. I need to see how the "withdrawal" in this case meets that standard.

Let me analyze the facts again. The Examiner's action in this case seems to be a rejection based on independent inventions, just like in *In re Haas*. So, it's likely a reviewable adverse decision under *Hengehold*.

Therefore, the District Court was wrong to dismiss the appeal. The "withdrawal" was a rejection, and the plaintiff had the right to appeal.

I need to make sure my analysis is clear and concise, citing the relevant case law and statutes.
```
<end_of_thought>
```
PER CURIAM. 1 This is a patent case which is subject to disposal on purely procedural grounds. Following a practice known as 'withdrawal' which and gained some standing in the Patent Examiner's office, the Examiner rejected claims 12 to 17 of plaintiff-appellant's patent application and then 'withdrew' claims 23 and 24 from consideration. The effect of this was to leave plaintiff free to refile these claims separately. As of the time of the Examiner's action, however the 'withdrawal' was conceived in the Patent Office not to represent a rejection. Hence no appeal could be taken from the Examiner's order. 2 Plaintiff-appellant did, however, unsuccessfully petition the defendant Commissioner to review the Examiner's decision. It then sought to appeal to the District Court under 5 U.S.C. 702 (1970), and 28 U.S.C. 1361 (1970). The District Judge Relying on lack of finality of the 'withdrawal' procedure denied appellant's motion for summary judgment and dismissed the action. 3 Some time thereafter, on November 1, 1973, the Court of Customs and Patent Appeals decided a case called In re Haas, 486 F.2d 1053 (1973). The court held that when an Examiner designated an adverse decision as a 'withdrawal' of a claim from further consideration it was 'in fact a rejection of that claim on the ground that it encompasses independent and distinct inventions.' Id. at 1056. The court thereupon held that appeals could be taken from such a rejection. 4 'We find that the action taken by the examiner did in fact amount to a rejection of claims 1 and 2. Those claims were withdrawn from consideration not only in this application but prospectively in any subsequent application because of their content. In effect there had been a denial of patentability of the claims. Presumably only by dividing the subject matter into separate, and thus different, claims in plural applications could an examination of the patentability of their subject matter be obtained. Conclusion 5 'We are thus led inevitably to the conclusion that the examiner's action here meets the standard established in Hengehold for adverse decisions reviewable by the board.

Figure 14: Example of synthetic data (Law domain) with hidden thoughts

```
<start_of_thought>
OK, I need to convince the court that this toll bridge is a public work. What does that even mean?
Right, a public work is something for the common good, something the public has a right to use.
Hmm, how can I prove that? Maybe I can focus on the bridge's purpose. It's clearly built for people to
cross the river, not just as a fancy decoration.
Wait, that's not enough. I need to show it's more than just a private enterprise.
Aha! I can compare it to other things the public considers public works, like turnpikes and railroads. Even
though they charge tolls, they're still considered public because they serve a public purpose.
That's it! I can argue that the bridge, like those, is a public work even though it charges a toll. It's a public
right to use it, and the toll is just compensation for maintaining it.
I'll start by emphasizing the bridge's purpose: it's for everyone, not just a select few. Then, I'll draw the
analogy to turnpikes and railroads, highlighting how they're both public works despite charging tolls.
I'll also need to back this up with legal precedent, like that case from Comyns's Digest. That should make
my argument even stronger.
<end_of_thought>
```
JUSTICE BRADLEY, after stating the case, delivered the opinion of the court. In approaching the solution of the questions presented by this certificate, the first inquiry that naturally presents itself is, whether a toll-bridge like that referred to is a public bridge, and hence a work of internal improvement. And we can hardly refrain from expressing surprise that there should be any doubt on the subject. What was the bridge built for, if not fit for public use? Certainly not for the mere purpose of spanning the Platte River as an architectural ornament, however beautiful it may be as a work of art; nor for the private use of the common council and their families; nor even for the exclusive use of the citizens of Fremont. All persons, of whatever place, condition, or quality, are entitled to use it as a public thoroughfare for crossing the river. The fact that they are required to pay toll for its use does not affect the question in the slightest degree. Turnpikes are public highways, notwithstanding the exaction of toll for passing on them. Railroads are public highways, and are the only works of internal improvement specially named in the act; yet no one can travel on them without paying toll. Railroads, turnpikes, bridges, ferries, are all things of public concern, and the right to erect them is a public right. If it be conceded to a private individual or corporation, it is conceded as a public franchise; and the right to take toll is granted as a compensation for erecting the work and relieving the public treasury from the burden thereof. Those who have such franchises are agents of the public. They have, it is true, a private interest in the tolls; but the works are public, and subject to public regulation, and the entire public has the right to use them. These principles are so elementary in the common law, that we can hardly open our books without seeing them recognized or illustrated. Comyns's Digest, title "Toll-thorough," commences thus: "Toll-thorough is a sum demanded for a passage through an highway; or, for a passage over a ferry, bridge, &c.; or, for goods which pass by such a port in a river: and it may be demanded in consideration of the repair of the pavement in a high street; or, of the repair of a sea-wall, bridge, &c.; cleansing of a river, &c. But toll-thorough cannot be claimed simply, without any consideration.

Figure 15: Example of synthetic data (Law domain) with hidden thoughts

```
<start_of_thought>
```
OK, I need to write a legal argument for reversing the district court's decision on the design patent.
First, I need to understand the basis for the district court's ruling. They said the design was obvious. Right, obviousness in design patent cases means the design is not sufficiently different from prior art.
What are the elements for proving obviousness? I remember it involves a "primary reference" that is visually similar to the patented design. Wait, did the district court even find a primary reference?
Let me check the record. Ah, there's nothing in the record that visually resembles Durling's design. This is a strong point for reversal.
The court also mentioned some other designs, the Gentry and Gaines models. I need to analyze those.
Hmm, the Gaines model is similar to Schweiger, but the difference in the end tables doesn't matter for obviousness. The focus is on the overall visual impression. The Gaines model is essentially identical to Schweiger, so I can argue that it's cumulative.
The Gentry model is completely different. It's a linear sofa, not the multi-component, right-angled design of Durling. And the front rail is different too.
I can argue that neither the Gaines nor the Gentry models create basically the same visual impression as Durling's design.
This is a strong argument for reversal. I should emphasize the lack of a primary reference and the visual dissimilarity of the cited prior art.
```
<end_of_thought>
```
IV 15 In conclusion, the district court's verbal description reveals that it improperly interpreted Durling's claimed design in terms of design concepts rather than its visual appearance as a whole. The record contains no prior art design that creates basically the same visual impression as does Durling's claimed design. No primary reference has been shown to exist. Without such a primary reference, it is improper to invalidate a design patent on grounds of obviousness. The judgment of the district court must be 16 REVERSED. 1 In its opinion, the district court also mentions the Gentry model and a sectional sofa group manufactured by Gaines Manufacturing Company (the Gaines model). The only apparent difference between the Gaines and Schweiger models is that the end tables in the Gaines model are separate from the sofa whereas those in Schweiger are integrated. The appearance of the two, however, is essentially identical. Because the obviousness inquiry focuses upon appearance and not functional considerations, the Gaines model is cumulative to Schweiger, and, accordingly, we will refer only to the Schweiger model 2 When properly done, this verbal description should evoke the visual image of the design 3 Similarly, the Gentry model does not create basically the same visual impression as Durling's design. The Gentry model appears to consist of a single, linear sofa, unlike the multiple-component right-angled design of Durling. Moreover, the front rail in the Gentry model barely curves upward and provides almost total vertical support for the end table. The Gentry model therefore does not duplicate Durling's design of an upward sweeping front rail having very little vertical support

Figure 16: Example of synthetic data (Law domain) with hidden thoughts

<start_of_thought>
OK, I need to write a response to Coleman's claim that the court has made a mistake by confusing him with someone else.

First, I need to understand his argument. He's saying the court used the wrong Presentence Report and sentenced him based on someone else's case. That's a serious allegation!

I need to find evidence to refute this. I should start by looking at the record in all of Coleman's cases. Is there any indication that the court has confused him with anyone else?

Wait, I remember that the court has a duty to ensure the accuracy of sentencing information. They wouldn't just use the wrong PSR without noticing.

Let me check if there's any record of a "Vernon Carl Coleman" in this district. If there isn't, that would strongly suggest Coleman is mistaken.

Good, the record shows no other defendant with that name. That's a good start.

Now, I need to address his claim about the sentencing guidelines. I need to make sure the court didn't use any other defendant's convictions to enhance his sentence.

The record confirms that no other defendant's convictions were used in Coleman's sentencing.

I can now confidently state that the court has not confused Coleman with anyone else and has used the correct PSR in his sentencing. I should clearly and concisely present this evidence in the response.
<end_of_thought>

In his Reasons Restrictions Should Not Be Imposed, Coleman claims that the Court has confused him with someone named Vernon Carl Coleman and has mistakenly sentenced him using someone else's Presentence Report ("PSR") and criminal case. The Court has conducted a thorough examination of the record in all of Coleman's criminal cases and finds no evidence that the Court has confused Coleman with anyone else. There is no record of prosecution of anyone named Vernon Carl Coleman in this District. . . . The record further establishes that no other defendant's convictions have been used to enhance his sentences or calculate his sentencing range under the Sentencing Guidelines. . . .

Figure 17: Example of synthetic data (Law domain) with hidden thoughts

## G  PROMPTS

### G.1  PROMPT FOR GENERATING HIDDEN THOUGHTS

This prompt was used with LLMs to generate synthetic hidden thoughts. Importantly, the same prompt was applied to both STEM and Law corpora without any domain-specific customization. Therefore, the comparison between domains is fair, as both rely on the identical prompt template.

```
You are an AI assistant emulating the internal thought process
↪  of an expert preparing to write a technical text.

### Instructions:
1. Start with a clear goal.
   - Begin with: `"OK, [task]"`, where `[task]` is a concise
   ↪  statement of the objective.
   - Example: `"OK, let's derive the binomial theorem
   ↪  expansion formula."`

2. Recall essential knowledge.
   - Think aloud about formulas, definitions, theorems, or
   ↪  algorithms relevant to the task.
   - Use natural introspective phrases such as:
     - *"Wait, what was that formula again?"*
     - *"Right, I remember that..."*
     - *"Let me think about how this works..."*

3. Consider alternative approaches (2-3 options).
   - Analyze different methods to approach the problem.
   - Example:
     - *"I could derive this using combinatorial arguments or
     ↪  through mathematical induction."*
     - *"Should I use a direct implementation or incorporate
     ↪  error handling?"*

4. Evaluate these approaches through reasoning.
   - Work through each method briefly, showing your
   ↪  calculations when appropriate.
   - Express uncertainty, reconsideration, or realization in
   ↪  natural language:
     - *"Hmm, that doesn't quite work because..."*
     - *"Actually, let me try a different approach..."*
     - *"Wait, I see a simpler way to do this..."*

5. Select and develop the most appropriate approach.
   - Choose the method that best aligns with the given text.
   - Show your work or derivation in detail, including:
     - Step-by-step calculations
     - Edge case analysis
     - Theoretical justifications

6. Verify your solution.
   - Check for correctness, especially for mathematical
   ↪  derivations or code implementations.
   - Consider special cases or potential issues.
   - Example:
     - *"Let me double-check this formula..."*
     - *"What happens when the input is zero or approaches
     ↪  infinity?"*
```

```
### Response Format
#### Analysis:
- What is the goal of this text?
  - [Goal description]
- What implicit background knowledge and omitted reasoning
↪   steps did the author likely possess?
  - [List specialized knowledge that the author likely
  ↪   possessed but did not explicitly mention in the text.
  ↪   Include exact formulas, theorems, algorithms,
  ↪   definitions, theoretical foundations, edge cases,
  ↪   implementation details, and domain-specific concepts
  ↪   that an expert would know]
- What approaches could be considered?
  - [List 1-2 potential approaches]
- Which approach should be chosen?
  - [Explain why the approach used in the text is most
  ↪   appropriate]

#### Thoughts:
<start_of_thought>
[Based on the analysis above, write a realistic internal
↪   thought process of the author, showing how they would
↪   think through creating the text]
<end_of_thought>

---

### Example1:
#### Text:
```
```
The binomial theorem states that $(a + b)^n$ can be expanded
↪   into a sum involving terms of the form $C(n, k) * a^{n-k}
↪   * b^k$. The coefficient $C(n, k)$ is defined as
↪   $\\frac{n!}{k!(n-k)!}$, where $n!$ represents the
↪   factorial of $n$. Each term in the expansion involves a
↪   combination of powers of $a$ and $b$ such that the sum of
↪   their exponents equals $n$.
```
```
#### Analysis:

- What is the goal of the explanation?
  - To derive the binomial theorem expansion formula.

- What knowledge needs to be recalled first?
  - The definition of the binomial coefficient $C(n, k)$.
  - How the coefficients in the expansion of $(a + b)^n$ are
  ↪   determined.

- What approaches could be considered?
  1. Combinatorial interpretation - Understanding each
  ↪   coefficient as a count of ways to choose terms in the
  ↪   expansion.
  2. Inductive proof - Assuming the theorem holds for $n$ and
  ↪   proving it for $n+1$.

- Which approach should be chosen?
```

```
    - Induction was considered first, but it is not immediately
    ↪  intuitive how the general form emerges.
    - The combinatorial approach makes it clearer why $C(n, k)$
    ↪  appears naturally in the expansion.

#### Thoughts:
<start_of_thought>
OK, let's derive the binomial theorem expansion formula.
Hmm, what's the best way to approach this? Wait, first, let me
↪  recall how the coefficients in $(a + b)^n$ are determined.
↪  Right, they come from choosing which terms get multiplied
↪  in the expansion. That means the coefficients correspond
↪  to selecting a certain number of $b$s from $n$ terms.
Wait, I remember now! That's exactly what the binomial
↪  coefficient $C(n, k)$ counts--the number of ways to choose
↪  $k$ $b$s out of $n$. So each term in the expansion must
↪  have a coefficient of $C(n, k)$. That makes sense.
What about an inductive proof? Hmm... if I assume it works for
↪  $n$, I need to show it works for $n+1$. Let me think...
↪  expanding $(a + b)^{n+1}$ using $(a + b)(a + b)^n$ should
↪  give me the next step, but the coefficients need careful
↪  handling.
Actually, that seems a bit messy. Tracking how the
↪  coefficients evolve through induction is less intuitive.
↪  Instead, thinking in terms of combinatorial counting gives
↪  a direct way to see why each term has the coefficient
↪  $C(n, k)$. OK, combinatorial arguments make the most sense
↪  here.
<end_of_thought>
---

### Example2:
#### Text:
```
{text}
```

#### Analysis:
```

## G.2  PROMPT FOR EVALUATION

For both MMLU and GPQA evaluations, we used the same evaluation prompt format. Specifically, we employed 2-shot prompts that include hidden thoughts enclosed by `<start_of_thought>` and `<end_of_thought>` tags. We intentionally limited the number of shots to two, since excessive few-shot examples may bias the model by pulling its reasoning process toward the chain-of-thoughts shown in the examples. Our goal here is to evaluate the inherent reasoning ability of the pretrained models rather than their ability to mimic the reasoning style of provided demonstrations. This setup ensures that differences in performance between benchmarks are not due to variations in prompt design but rather reflect the underlying reasoning capabilities of the models.

```
Problem 1: Emma has eight books. For her birthday, she got
 ↪  three books from her aunt and two books from her grandma.
 ↪  How many books does she have now? Options: A) 5 B) 8 C) 13
 ↪  D) 15
OK, let's solve this problem.

First, I need to figure out how many books Emma got in total.
 ↪  She got three from her aunt and two from her grandma, so
 ↪  that's a total of 3 + 2 = 5 new books.

Now, I need to add those new books to the books she already
 ↪  had. She started with 8 books, and then got 5 more. So, 8 +
 ↪  5 = 13.

Therefore, Emma has 13 books in total now.
<end_of_thought>
Answer: C

---

Problem 2: In criminal law, which legal principle states that
 ↪  a person cannot be tried twice for the same crime?
 ↪  Options: A) Habeas corpus B) Double jeopardy C)
 ↪  Self-incrimination D) Due process
<end_of_thought>
Hmm, this question is about a legal principle. I remember that
 ↪  double jeopardy is a constitutional protection against
 ↪  being tried twice for the same crime.

The other options are related to other legal concepts. Habeas
 ↪  corpus is about the right to challenge unlawful detention.
 ↪  Self-incrimination protects against being forced to
 ↪  testify against oneself. Due process ensures fair
 ↪  treatment under the law.

So the answer must be B) Double jeopardy.
<end_of_thought>
Answer: B

---

Problem3: {Question} Options: A) {A} B) {B} C) {C} D) {D}
<start_of_thought>
```

## G.3  PROMPT FOR CATEGORIZING THE DIFFICULTY OF MMLU QUESTIONS

```
You are an expert at evaluating the difficulty of academic
↪  problems. Your task is to analyze the given problem and
↪  classify its difficulty level on a scale of 1-5, where:

1: Very Easy
2: Easy
3: Medium
4: Hard
5: Very Hard

For each problem, provide only a difficulty rating (1-5).

---

Example (Very Easy):
Question: What is the sum of 5 + 7?
Options:
A) 10
B) 11
C) 12
D) 13
Answer: C

Difficulty: 1

---

Example (Easy):
Question: Solve for x: 2x + 3 = 11
Options:
A) x = 3
B) x = 4
C) x = 5
D) x = 6
Answer: B

Difficulty: 2

---

Example (Medium):
Question: If f(x) = 3x² - 2x + 1, find f'(2).
Options:
A) 8
B) 10
C) 12
D) 14
Answer: B

Difficulty: 3

---

Example (Hard):
Question: Let p = (1, 2, 5, 4)(2, 3) in S_5. Find the index of
↪  <p> in S_5.
Options:
```

```
A) 12
B) 20
C) 24
D) 30
Answer: C

Difficulty: 4

---

Example (Very Hard):
Question: Find the limit as x approaches infinity of (1 +
↪  1/x)^(x²) * (1 + 2/x)^(-x/2).
Options:
A) 0
B) 1
C) e
D) e^(-1)
Answer: D

Difficulty: 5

---

Now, please rate the difficulty of the following problem:

Question: {input_example}
Options:
A) {A}
B) {B}
C) {C}
D) {D}
Answer: {answer_example}

Difficulty:
```

### G.4 Prompt for 1-shot hidden thoughts on GSM8k

```
Problem 1: There are 15 trees in the grove. Grove workers will
↪  plant trees in the grove today. After they are done, there
↪  will be 21 trees. How many trees did the grove workers
↪  plant today?
<start_of_thought>
OK, let's solve this problem.
First, we know there were originally 15 trees in the grove.
↪  Now, after some trees were planted, the total count has
↪  increased to 21.
Hmm, what's the key operation here? Right, I need to figure
↪  out how many trees were added. If I take the final count
↪  and subtract the original count, that should give me the
↪  number of trees that were planted.
21 - 15... Let me double-check. Yes, that's 6. That makes
↪  sense--if we started with 15 and added 6 more, we'd get
↪  21.
OK, so the number of trees planted must be 6.
<end_of_thought>

Final Answer: 6

---

Problem 2: {question}
<start_of_thought>
```

### G.5 PROMPT FOR 5-SHOT CoT ON GSM8K

```
Q: There are 15 trees in the grove. Grove workers will plant
↪  trees in the grove today. After they are done, there will
↪  be 21 trees. How many trees did the grove workers plant
↪  today?
A:
<start_of_thought>
There are 15 trees originally.
Then there were 21 trees after some more were planted.
So there must have been 21 - 15 = 6.
<end_of_thought>
The answer is 6
---

Q: If there are 3 cars in the parking lot and 2 more cars
↪  arrive, how many cars are in the parking lot?
A:
<start_of_thought>
There are originally 3 cars.
2 more cars arrive. 3 + 2 = 5.
<end_of_thought>
The answer is 5
---

Q: Leah had 32 chocolates and her sister had 42. If they ate
↪  35, how many pieces do they have left in total?
A:
<start_of_thought>
Originally, Leah had 32 chocolates.
Her sister had 42.
So in total they had 32 + 42 = 74.
After eating 35, they had 74 - 35 = 39.
<end_of_thought>
The answer is 39
---

Q: Jason had 20 lollipops. He gave Denny some lollipops. Now
↪  Jason has 12 lollipops. How many lollipops did Jason give
↪  to Denny?
A:
<start_of_thought>
Jason started with 20 lollipops.
Then he had 12 after giving some to Denny.
So he gave Denny 20 - 12 = 8.
<end_of_thought>
The answer is 8
---

Q: Shawn has five toys. For Christmas, he got two toys each
↪  from his mom and dad. How many toys does he have now?
A:
<start_of_thought>
Shawn started with 5 toys.
If he got 2 toys each from his mom and dad, then that is 4
↪  more toys.
5 + 4 = 9.
<end_of_thought>
```

```
The answer is 9
---

Q: [[question]]
A:
<start_of_thought>
```

