# OpenReview forum: "Mining Hidden Thoughts from Texts: Evaluating Continual Pretraining with Synthetic Data for LLM Reasoning"
_ICLR.cc/2026/Conference — Submitted to ICLR 2026_

### Official Review · Reviewer_mtaB · 2025-10-28

**Soundness:** 2
**Presentation:** 3
**Contribution:** 2
**Rating:** 4
**Confidence:** 3

**Summary:**

This paper proposes Reasoning Continual Pretraining (**Reasoning CPT**), a method that enhances LLM reasoning by generating and incorporating hidden thoughts—synthetic representations of the implicit reasoning behind texts—into continual pretraining. Unlike SFT or RL approaches requiring task-specific supervision, Reasoning CPT leverages domain-general synthetic data to improve reasoning ability across diverse fields.

Experiments on MMLU and GPQA show consistent gains over standard CPT, with notable improvements on harder problems and strong cross-domain transfer: models trained on legal texts even outperform those trained on STEM data. These results suggest that reasoning skills can emerge from learning implicit cognitive processes, offering a scalable, domain-agnostic alternative to traditional reasoning model training.

**Strengths:**

1. The paper introduces the concept of **hidden thought trajectories** into continual pretraining, differing from RL and SFT approaches that rely on explicit reward signals. It proposes a **domain-agnostic mechanism** for generating reasoning data without human annotation or reward supervision, making the method broadly applicable across domains.

2. The study demonstrates that **reasoning ability can transfer from non-STEM texts**, such as legal documents, to STEM tasks. The experiments and analyses are thorough and well-supported, providing **theoretical insight** into how reasoning emerges in large language models.

**Weaknesses:**

1. The concept of **hidden thoughts** is described vaguely, lacking a formal definition and relying on heuristic prompt design rather than a principled formulation.
2. The **generation model (Gemma2-9B-it)** may limit quality, potentially introducing noise or fictitious reasoning traces that are not analyzed.
3. Evaluation is restricted to **MMLU and GPQA**, without testing other reasoning styles (e.g., CoT, multi-hop QA, code reasoning), and the paper does not isolate the effect of different hidden-thought components.
4. The study lacks direct comparison with SFT or post-training baselines, making the use of CPT as the only baseline relatively weak for assessing reasoning improvements.

**Questions:**

1. (Lines 078–083): Why are the proposed “hidden thoughts” considered *hidden* if they are explicitly generated as token sequences? How does this concept differ from existing definitions of latent or hidden rationales?
2. How do you ensure the quality and reliability of the hidden thoughts generated by **Gemma2-9B-it**, given its limited reasoning capability?
3. (Section 2.2): Have you compared Reasoning CPT with CPT using existing post-training datasets? How sensitive are the results to the choice of generator and sequence-length truncation?
4. (Lines 222–227): Since `<start_of_thought>` and `<end_of_thought>` are treated as normal tokens, how is reasoning completeness ensured during inference? Do you require special prompting?

---

> ### Author Response · Authors · 2025-12-02
> **Response to Reviewer mtaB (1/2)**
>
> Thank you very much for your recognition that we "propose a domain-agnostic mechanism" and that "experiments and analyses are thorough and well-supported."
>
> In response to your comments, we have clarified the definition of "hidden thoughts," provided considerations regarding comparison with existing post-training datasets, clarified the evaluation scope and future directions, and explained how reasoning completeness is ensured during inference.
>
> We believe these responses appropriately address the concerns you raised. We thank you again for the constructive feedback.
>
> ## W1: Definition of "Hidden Thoughts"
> **Response:**
>
> Thank you for pointing out the definition. We provide the following clarified definition:
>
> > *Hidden thoughts* represent the implicit reasoning processes that we assume underlie the creation of a text. Specifically, given a text $S$, we make the following assumptions: (1) there exists an author who wrote $S$, (2) the author engaged in internal reasoning processes while writing $S$ (such as recalling background knowledge, performing logical reasoning, making decisions, and self-verification), and (3) these reasoning processes are not explicitly written in $S$ itself.
> Based on these assumptions, we use an LLM to generate hidden thoughts $H$ that reconstruct these implicit reasoning processes: $H = \text{LLM}(S; P)$, where $P$ is a prompt designed to elicit such reasoning.
>
> We plan to include these in the updated version.
>
> ## W2 & Q2: How Do You Ensure Generation Quality?
> **Response:**
>
> Thank you for this sharp question. The thoughts generated by Gemma2-9B-it may contain noise. However, the final reasoning performance (MMLU, GPQA improvements) suggests that the generated thoughts are useful for training. In other words, this shows that reasoning performance improves with Reasoning CPT even with data containing noise.
>
> We used Gemma2-9B-it for generating hidden thoughts, but more powerful models (Gemini 3 Pro, GPT-5, DeepSeek-R1) could also be used.
>
> Also, evaluation of generation quality is an important direction, and we plan to include these explanations in the updated version.
>
> ## W3: Limited Evaluation Tasks, Insufficient Ablation
> **Response:**
>
> Thank you for your comment. Evaluation on other reasoning styles has been conducted in the paper's appendices. These analyses yielded the following findings:
>
> **Appendix B:** Pass@k analysis on GSM8k. Reasoning CPT surpasses Gemma2-9B-it's Pass@5 at just Pass@2 (91.7%), demonstrating the ability to generate diverse reasoning paths.
>
> **Appendix C:** Comparison of CoT style (5-shot) and Hidden Thoughts style (1-shot). Hidden Thoughts style consistently achieved higher accuracy (65.4% vs 58.3% for Gemma2-9B).
>
> **Evaluation Benchmark Selection:**
>
> We propose a training method for reasoning models on non-STEM text. Therefore, we selected evaluation benchmarks from the following perspectives: (1) A benchmark with very high difficulty used for evaluating reasoning models (GPQA), (2) A benchmark that can evaluate non-STEM QA (MMLU). While we have not conducted evaluations on multi-hop QA or code reasoning, we completely agree that these are important for evaluating such capabilities of models in the future. Therefore, we plan to include descriptions of these as important future directions in the updated version of the paper.
>
> ## W4: No Comparison with SFT/Post-training
>
> We conducted comparisons with the method (SFT) used in training the state-of-the-art reasoning model s1  [Muennighoff+2025]. Please refer to our response in "General Response to Common Concerns."
>
> ## Q1: Why "Hidden" When Explicitly Generated?
>
> **Why "hidden":** We assume there is an author who wrote the text, and that the text was created through the author's aforementioned internal reasoning (this need not be fact but is an assumption for creating synthetic data). Since the thoughts existed but are not explicitly written in the text, we call them hidden thoughts. We generate hidden thoughts through LLM generation.
>
> ## Q3: CPT Training on Post-training Datasets
> **Response:**
>
> **Regarding comparison with existing post-training datasets:**
>
> In the experiments shown in our response to "General Response to Common Concerns (Comparison with SFT)," we conducted SFT using long CoT generated by Gemini. As a result, the combination of Reasoning CPT + SFT showed the best performance. This result suggests that the Hidden Thoughts approach is complementary to existing high-quality reasoning data.
>
> **Regarding sensitivity to generator choice and sequence-length truncation:**
>
> In our current experiments, we use Gemma2-9B-it as the generation model and limit the length of Hidden Thoughts to 512 tokens. Systematic comparison using different generation models (e.g., more powerful models or different architectures) or different length limits is a future research topic that we plan to include in the updated version.

---

> > ### Author Response · Authors · 2025-12-02
> > **Response to Reviewer mtaB (2/2)**
> >
> > ## Q4: Reasoning Completeness During Inference
> > **Response:**
> >
> > As you pointed out, `<start_of_thought>` and `<end_of_thought>` are encoded as normal tokens using existing subword tokenization, not as new special tokens (Section 2.3).
> >
> > During inference, we use a special prompt format that includes these thought tags. Specifically, as shown in Appendix G.2, we present examples of Hidden Thoughts enclosed by `<start_of_thought>` and `<end_of_thought>` tags in 2-shot prompts. This guides the model to generate thoughts in a similar format during inference.

---

### Official Review · Reviewer_kkHq · 2025-10-28

**Soundness:** 3
**Presentation:** 4
**Contribution:** 3
**Rating:** 8
**Confidence:** 4

**Summary:**

This work explores continual pre-training language models on documents enhanced by another language models. The enhancement is to generate "hidden thoughts" for the document, basically asking the LM to explain what essential knowledge is needed to write the document, other ways this document could've been written, validating the answers in the document etc. The authors focus on two subdomains for sourcing more pretraining data, Law (from the Pile) and STEM (from OpenWebMath) and use Gemma2-9B-it to create the hidden thoughts while training Gemma2-9B and Qwen2.5-7B. They find that by creating these hidden thoughts and prepending them to the document, you can continually pretrain these models and have them perform better on MMLU and GPQA over the base model and continual pretraining over just the documents themselves.

The authors further show that this is not a benefit from more tokens, and that a lot of the benefit comes from being able to solve challenging questions within their evaluation datasets (ones that require significant reasoning), and that a key feature for effective "hidden thoughts" comes from the length of the document (at least for the law domain).  The results also suggest some interesting cross-domain transfer, i.e. training on Law data transfers well to GPQA which requires math and science reasoning.  All of this results in a method that can be used to synthetically enhance existing pretraining datasets, encouraging the models to reason more effectively.

Overall, I liked the work! I think this seems pretty easy to implement with nice findings and is well written.

**Strengths:**

- **The results are quite promising.** The hidden thoughts on the Law domain seem to really aid the downstream performance in somewhat surprising ways, i.e., enhancing the performance on GPQA for Qwen2.5-7B
- **Good reproducability** The process is very straightforward and well documented for other researchers/practitioners to use and experiment on.
- **Very clear, nice figures & tables, and well written.** The tables and figures present clear results and the paper itself is structured nicely (good flow).

**Weaknesses:**

- **Weak analysis**, more time could be spent on figuring out where these gains are coming from. For example, "To CoT or not to CoT? Chain-of-thought helps mainly on math and symbolic reasoning" is a paper that shows most of the gain on MMLU comes from questions that are math-heavy (they find it by looking for "=" in the generated responses because math questions are sometimes placed into unassuming categories like "business").  I would be interested in seeing if some of the cross-domain transfer we see for MMLU and GPQA for that matter are derived from this (math questions in seemingly unrelated math domains).  Additionally, the prompt used to generate hidden thoughts seems based off of what I typically see in "R1" style reasoning. I wonder if this format (alternative methods / verify / etc.) is just well suited for math topics and hasn't actually trained the model to "reason" on other domains where this wouldn't help.  I like the length experiment, but I would like to see more qualitative analysis and thought put into what these hidden thoughts are really teaching the model and how far we think it can extend.
- **Narrow scope**, a bit related to my previous remark, law is often tightly coupled with "logic" and STEM is often tied with math, the intro kind of makes broad statements about how this could be applied to other domains we typically do not see reasoning working well in, but the domains evaluated and trained on are just that.  (Again MMLU is a bit nuanced because of the findings in that previous paper).  I would like to see evaluations OR training on domains that are less math and logic OR maybe an argument why STEM and Law paired with MMLU and GPQA are fine to generalize from (but I think this will be hard to make convincing, GPQA is also very very math heavy when you look at it).

One other paper that may be worth mentioning in the related work is "RATIONALYST: Mining Implicit Rationales for Process Supervision of Reasoning" by Jiang et al. they also "mine" thoughts from pretraining datasets (they do a slightly different training based on that data, but the mining of "thoughts" from pretraining data reminded me a lot of this work).

**Questions:**

See the weakness points, I inlined many questions there. I think it boils down to the generalizability of the method and findings. For example:
- Are we sure the gains we see in MMLU that are "cross-domain" are actually not math questions in disguise
- What would happen if you selected random pretraining documents to enhance with "hidden thoughts" or if you merged law and stem? (the answer to this is kind of telling for general practice, it tells us if we need to enhance subsets of our datasets or if we could just enhance any and every document we come across).
- What do the hidden thoughts look like for Law / STEM trained models when they are answering a question that is about "Humanities" in MMLU? Some analysis / statistics on the surface patterns for these could help argue for generalization (like are "non-math" questions in MMLU typically producing shorter hidden thoughts, less helpful hidden thoughts, etc.)

---

> ### Author Response · Authors · 2025-12-02
> **Response to Reviewer kkHq (1/4)**
>
> Thank you for your positive evaluation and for recognizing this work as having 'nice findings' that are 'quite promising.' We appreciate your comments on the reproducibility, clarity, and structure of our paper.
> Your comments provided a valuable opportunity to further strengthen this research. We have sincerely addressed each point by adding detailed analysis of cross-domain gains, adding RATIONALYST to Related Work, and responding to questions regarding data selection.
>
> We believe these responses appropriately address the points you raised. We sincerely thank you again for your support of this research and constructive feedback.
>
> ## W1 & W2 & Q1: Weak Analysis of Cross-Domain Gains / Want to See Evaluation on Domains with Less Math and Logic / Are Cross-Domain Gains from Math Problems?
> **Response:**
>
> Thank you for this very sharp and important observation. The prior work you mentioned, "To CoT or not to CoT?", found that CoT's MMLU accuracy improvement comes from questions containing "=". This finding has important implications: the performance improvements we observed may actually be only improvements on math problems, not true cross-domain transfer.
>
> As you pointed out, math questions in MMLU are sometimes placed in seemingly unrelated categories like "business." Therefore, even if we claim that "training on the law domain improved non-STEM categories," it may actually be that only hidden math problems within those categories improved.
>
> Therefore, we conducted the following analysis:
> - For each subject in MMLU (more granular categories than STEM or Social Sciences), we examined two things:
> - **% of questions containing '='**: The proportion of examples containing "=" in the problem (the higher this is, the more math-focused the problem)
> - **Diff**: The difference in accuracy between Reasoning CPT and CPT (the larger this is, the more Reasoning CPT contributes to accuracy improvement)
>
> The results are as follows. The rows in this table are sorted by Diff value, meaning subjects at the top had the largest Reasoning CPT contribution.
>
> Looking at this, **we can see that Reasoning CPT contributes to many subjects that contain no "=" at all**. For example, Reasoning CPT trained on law improves CPT by nearly 10% on "jurisprudence" and "world_religions," which contain no "=" (i.e., are not math-focused categories). Also, despite being trained on law, it improves by nearly 12% on "high_school_mathematics" and "college_computer_science." This shows that Reasoning CPT brings performance improvements not only to math-focused problems but also to various non-STEM categories.
>
> Your feedback has significantly strengthened our analysis. Thank you! We hope this addresses your concerns.

---

> > ### Author Response · Authors · 2025-12-02
> > **Response to Reviewer kkHq (2/4)**
> >
> > Table 1: MMLU subject-level analysis: Law domain CPT
> >
> >
> > |Category|Subject|(A) Acc. of ReasoningCPT| (B) Acc.of CPT| Diff (A-B)| % o fquestions containing '='|
> > |:----------------|:------------------------------------|--------------:|--------------:|----------:|----------------:|
> > |stem|college_computer_science|72.0|59.0|13.0|12.0|
> > |stem|high_school_mathematics|80.7|68.5|12.2|28.9|
> > |humanities|jurisprudence|79.6|69.4|10.2|0.0|
> > |humanities|world_religions|81.9|72.5|9.4|0.0|
> > |other|medical_genetics|74.0|67.0|7.0|0.0|
> > |other|business_ethics|74.0|67.0|7.0|0.0|
> > |stem|computer_security|82.0|76.0|6.0|3.0|
> > |stem|college_biology|78.5|72.9|5.6|0.0|
> > |stem|electrical_engineering|65.5|60.0|5.5|0.0|
> > |other|marketing|87.6|82.1|5.5|0.0|
> > |social_sciences|human_sexuality|74.0|68.7|5.3|0.0|
> > |other|management|85.4|80.6|4.8|0.0|
> > |social_sciences|high_school_government_and_politics|87.6|83.4|4.2|0.0|
> > |social_sciences|sociology|76.1|72.1|4.0|0.0|
> > |stem|college_physics|64.7|60.8|3.9|9.8|
> > |other|anatomy|67.4|63.7|3.7|0.0|
> > |other|virology|47.6|44.0|3.6|0.6|
> > |humanities|high_school_world_history|75.5|72.2|3.3|0.0|
> > |humanities|professional_law|46.2|43.2|3.0|0.0|
> > |social_sciences|professional_psychology|65.8|63.1|2.7|0.2|
> > |stem|high_school_physics|62.9|60.3|2.6|8.6|
> > |humanities|moral_scenarios|47.0|44.5|2.5|0.0|
> > |humanities|formal_logic|49.2|46.8|2.4|0.0|
> > |social_sciences|high_school_microeconomics|83.2|81.1|2.1|0.8|
> > |other|college_medicine|66.5|64.7|1.8|0.6|
> > |social_sciences|public_relations|69.1|67.3|1.8|0.0|
> > |social_sciences|high_school_psychology|86.1|84.4|1.7|0.0|
> > |social_sciences|high_school_geography|84.8|83.3|1.5|0.0|
> > |stem|high_school_biology|81.6|80.3|1.3|0.3|
> > |stem|elementary_mathematics|91.8|90.5|1.3|11.9|
> > |other|miscellaneous|85.1|84.0|1.1|0.0|
> > |stem|college_mathematics|55.0|54.0|1.0|38.0|
> > |social_sciences|us_foreign_policy|81.0|80.0|1.0|0.0|
> > |humanities|philosophy|66.9|66.2|0.7|0.0|
> > |humanities|moral_disputes|63.0|62.7|0.3|0.0|
> > |stem|abstract_algebra|57.0|57.0|0.0|27.0|
> > |social_sciences|high_school_macroeconomics|73.6|73.6|0.0|0.5|
> > |other|professional_accounting|54.6|54.6|0.0|0.0|
> > |stem|high_school_computer_science|85.0|85.0|0.0|20.0|
> > |social_sciences|security_studies|62.4|62.9|-0.5|0.0|
> > |other|nutrition|70.9|71.6|-0.7|0.0|
> > |stem|high_school_statistics|70.8|71.8|-1.0|12.5|
> > |humanities|logical_fallacies|72.4|73.6|-1.2|0.0|
> > |stem|conceptual_physics|74.0|75.3|-1.3|0.4|
> > |stem|astronomy|71.7|73.0|-1.3|0.0|
> > |stem|high_school_chemistry|57.6|59.1|-1.5|2.5|
> > |humanities|high_school_us_history|74.5|76.0|-1.5|0.0|
> > |social_sciences|econometrics|51.8|53.5|-1.7|7.0|
> > |stem|college_chemistry|46.0|48.0|-2.0|12.0|
> > |humanities|prehistory|70.1|72.2|-2.1|0.0|
> > |humanities|international_law|71.1|73.6|-2.5|0.0|
> > |other|human_aging|65.0|67.7|-2.7|0.0|
> > |humanities|high_school_european_history|72.7|75.8|-3.1|0.0|
> > |stem|machine_learning|50.0|53.6|-3.6|8.0|
> > |other|professional_medicine|68.8|72.8|-4.0|1.1|
> > |other|clinical_knowledge|71.3|76.6|-5.3|0.0|
> > |other|global_facts|38.0|49.0|-11.0|0.0|

---

> > > ### Author Response · Authors · 2025-12-02
> > > **Response to Reviewer kkHq (3/4)**
> > >
> > > Table 2: MMLU subject-level analysis: STEM domain CPT
> > >
> > > |Category|Subject|(A) Acc. of ReasoningCPT| (B) Acc.of CPT| Diff (A-B)| % o fquestions containing '='|
> > > |:----------------|:------------------------------------|--------------:|--------------:|----------:|----------------:|
> > > |humanities|world_religions|82.5|70.2|12.3|0.0|
> > > |social_sciences|us_foreign_policy|81.0|70.0|11.0|0.0|
> > > |other|anatomy|65.9|56.3|9.6|0.0|
> > > |stem|high_school_computer_science|89.0|80.0|9.0|20.0|
> > > |other|business_ethics|71.0|63.0|8.0|0.0|
> > > |other|medical_genetics|80.0|73.0|7.0|0.0|
> > > |stem|high_school_mathematics|76.3|70.4|5.9|28.9|
> > > |humanities|logical_fallacies|76.7|71.2|5.5|0.0|
> > > |stem|college_biology|79.2|74.3|4.9|0.0|
> > > |other|professional_accounting|59.9|55.0|4.9|0.0|
> > > |stem|astronomy|78.3|73.7|4.6|0.0|
> > > |stem|computer_security|78.0|74.0|4.0|3.0|
> > > |stem|machine_learning|58.0|54.5|3.5|8.0|
> > > |social_sciences|high_school_government_and_politics|83.9|81.3|2.6|0.0|
> > > |humanities|high_school_world_history|76.4|73.8|2.6|0.0|
> > > |social_sciences|econometrics|54.4|51.8|2.6|7.0|
> > > |other|virology|50.6|48.2|2.4|0.6|
> > > |social_sciences|human_sexuality|72.5|70.2|2.3|0.0|
> > > |social_sciences|security_studies|67.3|65.3|2.0|0.0|
> > > |social_sciences|professional_psychology|65.7|63.7|2.0|0.2|
> > > |other|professional_medicine|69.5|67.6|1.9|1.1|
> > > |other|nutrition|70.9|69.0|1.9|0.0|
> > > |stem|conceptual_physics|72.3|71.1|1.2|0.4|
> > > |stem|college_computer_science|66.0|65.0|1.0|12.0|
> > > |stem|high_school_biology|78.7|77.7|1.0|0.3|
> > > |humanities|formal_logic|50.0|49.2|0.8|0.0|
> > > |stem|electrical_engineering|66.9|66.2|0.7|0.0|
> > > |humanities|professional_law|44.3|43.6|0.7|0.0|
> > > |social_sciences|high_school_geography|83.3|82.8|0.5|0.0|
> > > |social_sciences|sociology|73.6|73.1|0.5|0.0|
> > > |stem|high_school_statistics|69.9|69.4|0.5|12.5|
> > > |social_sciences|high_school_macroeconomics|74.9|74.4|0.5|0.5|
> > > |other|miscellaneous|82.4|82.0|0.4|0.0|
> > > |social_sciences|high_school_microeconomics|81.5|81.1|0.4|0.8|
> > > |stem|elementary_mathematics|89.4|89.2|0.2|11.9|
> > > |stem|college_physics|62.7|62.7|0.0|9.8|
> > > |other|college_medicine|65.3|65.3|0.0|0.6|
> > > |other|management|77.7|77.7|0.0|0.0|
> > > |humanities|prehistory|69.1|69.4|-0.3|0.0|
> > > |other|clinical_knowledge|70.6|70.9|-0.3|0.0|
> > > |other|marketing|84.2|85.0|-0.8|0.0|
> > > |humanities|international_law|68.6|69.4|-0.8|0.0|
> > > |humanities|jurisprudence|70.4|71.3|-0.9|0.0|
> > > |stem|high_school_physics|57.6|58.9|-1.3|8.6|
> > > |social_sciences|high_school_psychology|84.4|86.1|-1.7|0.0|
> > > |social_sciences|public_relations|60.0|61.8|-1.8|0.0|
> > > |humanities|moral_scenarios|45.5|47.6|-2.1|0.0|
> > > |stem|high_school_chemistry|64.0|66.5|-2.5|2.5|
> > > |other|human_aging|63.7|66.8|-3.1|0.0|
> > > |humanities|philosophy|66.9|70.1|-3.2|0.0|
> > > |humanities|high_school_us_history|72.1|75.5|-3.4|0.0|
> > > |humanities|moral_disputes|62.7|66.2|-3.5|0.0|
> > > |stem|college_chemistry|42.0|46.0|-4.0|12.0|
> > > |stem|abstract_algebra|49.0|53.0|-4.0|27.0|
> > > |humanities|high_school_european_history|72.7|77.6|-4.9|0.0|
> > > |other|global_facts|43.0|50.0|-7.0|0.0|
> > > |stem|college_mathematics|48.0|59.0|-11.0|38.0|

---

> > > > ### Author Response · Authors · 2025-12-02
> > > > **Response to Reviewer kkHq (4/4)**
> > > >
> > > > ## W3: Paper Worth Mentioning in Related Work
> > > > **Response:**
> > > >
> > > > Thank you for pointing out this valuable reference. Jiang et al.'s paper is an important related work that we had missed.
> > > >
> > > > As you noted, RATIONALYST also shares a similar idea with our method in mining hidden information from text. Specifically, they use LLMs to extract and filter implicit rationales hidden in text (e.g., rule violations lead to punishment), and use approximately 79k of these to train a specialized model that generates implicit rationales.
> > > >
> > > > However, there are differences in training objectives and usage methods. RATIONALYST trains a verification model for process supervision and uses it as guidance for selecting next-step candidates during inference. In contrast, our Reasoning CPT performs continual pre-training on data augmented with Hidden Thoughts, strengthening the reasoning foundation of the model itself.
> > > >
> > > > In any case, it is an important work that shares a common approach of mining information that supports reasoning from pre-training data, and we plan to add it to Related Work in the updated version. Thank you again for bringing this important reference to our attention.
> > > >
> > > > ## Q2: Domain Mixing Experiments
> > > > **Response:**
> > > >
> > > > Randomly selecting documents from different domains is a direction we are quite interested in and has high value to explore. However, unfortunately, we did not have enough time for continual pre-training and data construction to conduct these experiments. We plan to include an explanation in the updated paper as an interesting future research direction.
> > > >
> > > > Instead, we did conduct experiments on "Training with Mixed Domains," so please refer to the response in "General Response to Common Concerns."

---

### Official Review · Reviewer_pwXB · 2025-10-31

**Soundness:** 2
**Presentation:** 2
**Contribution:** 2
**Rating:** 2
**Confidence:** 4

**Summary:**

This paper propose reasoning continual pre-training (Reasoning CPT), a form of CPT that uses synthetic data to generate the hidden thought processes underlying texts, based on the premise that texts are the result of the author’s thinking process. Analysis shows that Reasoning CPT can significantly enhance reasoning ability even when trained on non-STEM corpora that have rarely been used for reasoning tasks. On both MMLU and GPQA, Reasoning CPT achieved substantial improvements over the base model and standard CPT.

**Strengths:**

1. The hidden thoughts are effective for reasoning tasks.
2. The mechanism behind hidden thoughts are transferable.

**Weaknesses:**

1. The ``hidden thoughts'' is similar to generating chain-of-thought or slow thinking process for a piece of pre-training text.
2. Insufficient experiments.
3. The analyses of synthetic data provide neither in-depth explanation for the proposed method nor useful insights for future directions.

details can refer to the questions below.

**Questions:**

1. What are the differences among chain-of-thought, slow thinking process in o1 or deepseek-r1, and your proposed hidden thought? I know hidden thought is for pre-training text, but I do not think they have fundamental differences. If they are similar, then hidden thoughts is not novel. If they are not similar, then you should have comparisons among them.
2. Reasoning CPT works in both Law and STEM domains, and it is transferable, why not combine data in Law and STEM for experiments? I should have further improvement.
3. This paper focus on continual pre-training, while the data size is small. Moreover, the settings of CPT have some problems such as ``6 epoches'' and "learning rate of 3e-5". Does pre-training requires multiple runs of training? If they are the consensus in CPT research area, then I would take back this question.
4. What is the purpose to conduct the analyses in section 4? I think analyses should give in-depth investigations or explanation to prove the effectiveness of your method, or give insights for future diorections.

---

> ### Author Response · Authors · 2025-12-02
> **Response to Reviewer pwXB (1/2)**
>
> Thank you very much for your highly constructive feedback. We appreciate your recognition of the effectiveness and transferability of Hidden Thoughts.
>
> In response to your comments, we have clarified the fundamental differences from CoT/slow thinking processes, added domain mixing experiments, clarified the purpose of the analysis in Section 4, and explained the validity of CPT settings.
>
> We believe these responses appropriately address the concerns you raised. We thank you again for the constructive feedback.
>
> ## W1 & Q1: Differences from CoT and Slow Thinking
> **Response:**
>
> We conducted comparisons with the method (SFT) used in training the state-of-the-art reasoning model s1  [Muennighoff+2025]. Please refer to our response in "General Response to Common Concerns."
>
> ## W2 & Q2: Experiments Combining Law and STEM Data
>
> **Response:**
>
> Thank you for this important point. Domain combination is important for validating the versatility of this method, and we completely agree with your opinion. Therefore, we conducted domain mixing experiments. For the results, please refer to "Training with Mixed Domains." We thank you for providing this valuable opportunity to improve the quality of this research!
>
> ## W3 & Q4: Purpose of Analysis in Section 4
> **Response:**
>
> Thank you for the constructive feedback. We clarify the purpose of the analysis in Section 4.
>
> The purpose of our analysis is to understand data characteristics and provide insights for future data construction.
>
> An important contribution of Reasoning CPT is the data method. We use LLMs to infer Hidden Thoughts behind raw text and use them as training data for continual pre-training. While Reasoning CPT showed good performance on benchmarks, we sought to understand why it is effective through data analysis.
>
> **Analytical Perspective:**
>
> Long thought processes are known to improve reasoning performance (e.g., s1, r1, and o1). In SFT and RL, LLMs generate long CoT according to problem difficulty, but there are limitations: (1) such data is limited, (2) domains are also limited to mathematics and programming.
>
> Our method solves these problems. However, when generating Hidden Thoughts, we did not instruct the LLM to generate long thoughts. Therefore, it was unclear whether we could train "long thoughts" like SFT and RL. Thus, we analyzed the correlation between the length of the original text and the length of the generated Hidden Thoughts.
>
> **Results:**
>
> As a result, strong positive correlations were observed (ρ=0.348 STEM, ρ=0.486 Law). This has important implications:
> - Long thought data can be easily created simply by preparing long texts, without worrying about constraints such as mathematical problems or their difficulty
> - Long thought data can be easily generated in domains where long texts exist abundantly, such as legal documents and scientific papers
>
> This finding demonstrates that the sources of long thought data can be created from far more domains than previously considered STEM domains. This provides practical insights that greatly expand domain choices for future data construction.
>
> We plan to make the explanation of Section 4 clearer according to this content in the updated version.

---

> > ### Author Response · Authors · 2025-12-02
> > **Response to Reviewer pwXB (2/2)**
> >
> > ## Q3: Validity of CPT Settings
> > **Response:**
> >
> > Thank you for this very sharp observation. We explain the validity of CPT settings.
> >
> > **(1) Multiple Epochs and Learning Rate**
> >
> > Multiple epochs and a learning rate around 1e-5 are common settings in CPT [Ke+2023] [Yang+2024].
> >
> > - [Ke+2023] Continual Pre-training of Language Models
> > - [Yang+2024] Synthetic continued pretraining
> >
> > **(2) Data Size**
> >
> > Regarding data size, we experimented with 36.8M-28.3M tokens, which is not the token count scale of full-scratch pre-training. We recognize that validation with larger-scale data would strengthen our findings and plan to note this in Limitations. However, in realistic continual pre-training in practice, there are very many situations where large amounts of data for specific domains are not available. Rather, we demonstrate that performance improvement can be achieved even with this size of data.
> >
> > **(3) 8× Data Efficiency**
> >
> > More importantly, data with Hidden Thoughts tends to converge very quickly in training within the same computational budget. In other words, Reasoning CPT has overwhelmingly better data efficiency than regular CPT.
> >
> > This is shown in the training loss trends in Figure 7. For example, in Gemma2-9B (Figure ( c)), to reach a loss of 1.25:
> > - Reasoning CPT: approximately 5M tokens (STEM & Law)
> > - Regular CPT: approximately 40M tokens (STEM), approximately 55M tokens (Law)
> >
> > are required. In other words, training with text augmented with Hidden Thoughts requires 1/8 to 1/11 of the computational budget. This demonstrates the superiority of Reasoning CPT over CPT in practical environments where only limited computational budgets or small amounts of domain-specific data are available.

---

### Official Review · Reviewer_nb3E · 2025-10-31

**Soundness:** 2
**Presentation:** 2
**Contribution:** 2
**Rating:** 4
**Confidence:** 3

**Summary:**

This paper proposes Reasoning Continual Pretraining, a method that enhances LLMs’ reasoning abilities by generating "hidden thoughts" (implicit expert thinking processes) behind texts to construct synthetic training data, and evaluates it across STEM and Law domains. It uses Gemma2-9B and Qwen2.5-7B as base models, comparing Reasoning CPT with base models and standard CPT on MMLU and GPQA benchmarks.

**Strengths:**

The detailed analysis of hidden thought characteristics (e.g., length correlation with original texts, domain variance) offers valuable insights into synthetic data construction for reasoning pretraining.

addresses a critical limitation of existing LLM reasoning training (over-reliance on task-specific signals) by validating non-STEM data’s effectiveness, providing a novel direction for data selection

**Weaknesses:**

- The abstract and introduction lack logical coherence. For instance, there is an abrupt transition from reasoning models to CPT in the abstract, with insufficient contextual connection to justify this shift.

 - No comparisons are made with other related state-of-the-art works. This omission prevents a clear demonstration of the proposed method’s advantages over existing approaches.

- The proposed method bears significant similarities to knowledge distillation directly from stronger reasoning models. Clarifying the essential differences between them is critical, and comparative performance evaluations against such distillation-based methods are indispensable to validate its uniqueness.

- The study does not conduct in-depth exploration of potential biases inherent in synthetic hidden thoughts, for example, domain-specific biases inherited from base models like Gemma2-9B-it. Such unaddressed biases may restrict the method’s practical applicability in fair and unbiased reasoning scenarios.

- The research is limited to only two domains (STEM and Law) without extending to other fields. As a result, it fails to fully corroborate the claimed cross-domain scalability of the proposed method.

**Questions:**

see Weaknesses

---

> ### Author Response · Authors · 2025-12-02
> **Response to Reviewer nb3E (1/2)**
>
> Thank you very much for your highly valuable feedback. We appreciate your recognition that the detailed analysis of Hidden Thoughts provides valuable insights for synthetic data construction, and that the validation of non-STEM data effectiveness indicates new directions for data selection.
>
> In response to your comments, we have improved the logical coherence of the abstract and introduction, added comparison experiments with related work, clarified differences from knowledge distillation, and conducted domain mixing experiments.
>
> We believe these responses appropriately address the concerns you raised. We thank you again for providing this valuable opportunity.
>
> ## W1: Lack of Logical Coherence in Abstract and Introduction
> **Response:**
>
> We will clarify the motivation for training reasoning models from the CPT stage as follows:
>
> **Previous reasoning model training:** SFT and RL have been adopted as training methods for reasoning models, but they have the following problems: (1) they require task-specific success signals, (2) therefore, they are limited to domains such as mathematics and programming where strict signals can be provided, and data cannot be effectively utilized.
>
> **CPT solves these problems:** (1) As self-supervised learning, it does not require success signals such as "correct" or "incorrect" for inputs, (2) Since it only requires text, it can be applied to non-STEM domains.
>
> Therefore, we train reasoning models using text from domains other than STEM from the CPT stage. However, the challenge is data for reasoning, and we propose an excellent data synthesis method for reasoning CPT.
>
> We plan to revise the abstract and introduction in the updated version.
>
> ## W2: Comparison with Related Work
> **Response:**
>
> We conducted comparisons with the method (SFT) used in training the state-of-the-art reasoning model s1. Please refer to our response in "General Response to Common Concerns."
>
> ## W3: Need to Clarify Similarities with Knowledge Distillation
> **Response:**
>
> Thank you for this sharp observation. We clarify the fundamental differences between knowledge distillation and Reasoning CPT.
>
> **General Approach of Knowledge Distillation:** Knowledge distillation is a method that trains student models on the logits or generated responses of teacher models. For example, in the DeepSeek-R1 paper, to give reasoning capabilities to small-scale models, they fine-tune small models using 800,000 samples prepared by DeepSeek-R1.
>
> **Fundamental Differences from Reasoning CPT:**
>
> 1. **Difference in Training Methods:** Distillation is generally SFT, not CPT. CPT and Reasoning CPT are self-supervised learning, so unlike SFT, they do not require teacher signals such as answers to instructions. Therefore, they can be easily trained using text from domains such as law.
>
> 2. **Difference in Objectives:** Reasoning CPT does not primarily aim to train small-scale models. In this paper, the model generating synthetic data and the model being trained are of comparable scale. In contrast, the main objective of distillation is to transfer knowledge from large-scale models to small-scale models.
>
> 3. **Difference in Source of Thought Induction:** Reasoning CPT adds thought processes to the text itself. Therefore, it differs from distillation in that thoughts are induced from pre-training text rather than from prompts.
>
> **Complementarity of Reasoning CPT and Distillation:** Furthermore, the SFT in Tables 1 and 2 uses long CoT generated from a frontier model (Gemini) for SFT, making it a setting close to distillation. Looking at these results, combining both Reasoning CPT and SFT improves performance. This indicates that distillation is not an opposing approach to Reasoning CPT, but rather orthogonal and complementary.
>
> We plan to include these explanations in the updated version.
>
> ## W4: Insufficient Bias Analysis
> **Response:**
>
> Thank you for pointing out concerns about LLM-inherent biases affecting synthetic data. As you mentioned, the Gemma2-9B-it we used may have some biases toward domains. For example, we created synthetic data in the law domain, but we cannot deny the possibility that this contains LLM-inherent political biases. Countermeasures include not creating synthetic data from texts that induce cultural or gender biases, and using models with fewer biases. However, this is an important concern common to methods that create synthetic data. Therefore, we consider it appropriate to include an explanation in the paper as an ethical concern. We plan to include these explanations in the updated version.

---

> > ### Author Response · Authors · 2025-12-02
> > **Response to Reviewer nb3E (2/2)**
> >
> > ## W5: Cross-Domain Scalability
> > **Response:**
> >
> > Thank you for this important point. We selected these two to compare STEM, the traditional training domain for reasoning models, with non-STEM domains. Legal data was selected as a representative example of non-STEM. Although legal texts exist in abundance, they have rarely been utilized for reasoning model training. We show that training by adding hidden thoughts to legal data achieves performance comparable to training on STEM alone. This is a new finding and we believe it is valuable information for model developers. We hope you will appreciate this point.
> >
> > To address your concerns, in addition to models trained on "STEM only" and "Law only," we newly conducted experiments training on a mixed domain combining both domains at 50% each. Please see "General Response to Common Concerns."

---

### Author Response · Authors · 2025-12-02
**General Response to Common Concerns (1/3)**

We sincerely thank all reviewers for their detailed feedback. Your comments have provided a valuable opportunity to significantly improve the quality of this research.

To appropriately address the major concerns, we have conducted comprehensive additional experiments. This response contains detailed experimental results and analyses that are planned to be incorporated into the paper (all of which will be fully reflected in the camera-ready version).

We believe these responses address the main concerns raised. Below are our responses to points commonly raised by multiple reviewers.

## Comparison with SFT

Similar concerns were raised by the following reviewers:
- nb3E (W2): Comparison with related work needed
- pwXB (W1 & Q1): Differences from CoT and slow thinking
- mtaB (W4): Lack of comparison with SFT/post-training

**Response**:
We clarify the fundamental differences between Reasoning CPT and reasoning training methods such as SFT and RL.

In short, Reasoning CPT is fundamentally different from post-training methods like SFT and RL in that training data creation is easier and the applicable domains are broader.

**1. SFT/RL Constraints: Limited to Mathematics and Programming**

Existing reasoning models such as s1 [Muennighoff+2025] are SFT-trained on mathematical problems with long chain-of-thought reasoning. Creating this training data with LLMs requires verifiable problem-answer pairs where correctness can be objectively determined. However, such verifiable pairs with long reasoning traces can only be prepared in large quantities for limited domains such as mathematics and programming where solutions can be automatically verified. As a result, the abundant legal documents and humanities texts that exist in the world could not be utilized as reasoning training data with long CoT.

 [Muennighoff+2025] s1: Simple test-time scaling

**2. Hidden Thought: Enabling Training Data Creation from Raw Text**

In contrast, Reasoning CPT performs CPT on data that augments pre-training text with thought processes (Hidden Thoughts). This has the following advantages:

| Aspect | Existing Methods (SFT/RL) | Reasoning CPT (Ours) |
|--------|---------------------------|----------------------|
| Training Data | Problem-answer pairs | Raw text only |
| Data Creation Method | Use only correct answers from problem-solving | Add **Hidden Thoughts** to text |
| Applicable Domains | Mathematics and programming only | Any text such as STEM or Law |

Since it is self-supervised learning, problem-answer pairs are not necessary. In other words, reasoning training data can be easily created even in non-mathematical domains simply by adding thoughts to existing high-quality text.

**3. Reasoning CPT and SFT are Orthogonal**

While existing approaches are dominated by methods like SFT, Reasoning CPT and SFT are orthogonal. That is, they are not in conflict with general SOTA methods and can be combined.

Nevertheless, we recognize the importance of comparison with SFT research using long CoT for reasoning model training, and we completely agree with the reviewers' comments. Therefore, to demonstrate the complementary relationship between Reasoning CPT and post-training methods, we conducted experiments to SFT models after CPT.

Preliminary Experiments: Research on the state-of-the-art reasoning model s1 [Muennighoff+2025] has shown that SFT with 1000 carefully curated long CoT examples significantly improves reasoning performance. Therefore, we performed SFT on Qwen2.5-7B models after CPT using this training data (s1k; 1000 long CoT examples).

However, these models trained with long CoT directly from the base model could not sufficiently acquire instruction-following abilities. Specifically, they could not generate anchors like "Final answer: A" for zero-shot prompts without input-output examples, and many repetitions of the same phrase occurred. This may be because s1 performs SFT on instruction-tuned models, whereas we directly performed SFT on base models (which lack instruction-following ability).

[Muennighoff+2025] s1: Simple test-time scaling

Experimental Setup: Therefore, we applied chat vectors [Huang+2023] of Qwen2.5-7B and Qwen2.5-7B-Instruct to all CPT models to endow instruction-following ability, and then performed SFT with s1k on those models. This allowed us to acquire instruction ability without training and resolved the previous issue.

[Huang+2023] Chat Vector: A Simple Approach to Equip LLMs with Instruction Following and Model Alignment in New Languages

---

> ### Author Response · Authors · 2025-12-02
> **General Response to Common Concerns (2/3)**
>
> **Results and Key Findings**:
>
> Synergistic Effect of Reasoning CPT + SFT: SFT on Reasoning CPT models showed the best results. This demonstrates that CPT and SFT are orthogonal and can improve LLM accuracy when combined.
>
> Contribution of This Research: We provide a complementary contribution that makes data from domains that SFT and RL could not utilize available for reasoning training. Through these experiments conducted in response to reviewer feedback, we were able to demonstrate that Reasoning CPT is not a competing approach to SFT, but rather a complementary foundation that enhances the effectiveness of existing post-training methods. We again thank you for the constructive feedback!
>
>
> **Table 1: GPQA**
> | Method | Domain | Biology | Chemistry | Physics | All |
> |--------|--------|---------|-----------|---------|-----|
> | Qwen2.5-7B | - | 26.3 | 20.4 | 30.2 | 25.2 |
> | + CPT | Law | 31.6 | 25.8 | 23.3 | 25.3 |
> | + CPT | STEM | 31.6 | 24.7 | 24.4 | 25.3 |
> | + CPT + SFT | Law | 42.1 | 26.9 | 36.0 | 32.3 |
> | + CPT + SFT | STEM | 36.8 | 25.8 | 38.4 | 32.3 |
> | + Reasoning CPT | Law | 26.3 | **31.2** | 30.2 | 30.3 |
> | + Reasoning CPT | STEM | 47.4 | 28.0 | 32.6 | 31.8 |
> | + Reasoning CPT + SFT | Law | 42.1 | 22.6 | **45.3** | 34.3 |
> | + Reasoning CPT + SFT | STEM | **57.9** | **31.2** | 38.4 | **36.9** |
>
> **Table 2: MMLU**
> | Method | Domain | STEM | Social Sciences | Humanities | Other | All |
> |--------|--------|------|-----------------|------------|-------|-----|
> | Qwen2.5-7B | - | 69.9 | 74.2 | 57.3 | 70.8 | 66.8 |
> | + CPT | Law | 70.3 | 73.3 | 56.0 | 71.3 | 66.4 |
> | + CPT | STEM | 70.4 | 73.4 | 57.0 | 69.8 | 66.5 |
> | + CPT + SFT | Law | 76.3 | 68.5 | 56.3 | 64.2 | 65.2 |
> | + CPT + SFT | STEM | 76.4 | 73.5 | 55.8 | 71.2 | 67.7 |
> | + Reasoning CPT | Law | 72.7 | **75.2** | 57.9 | 71.6 | 68.0 |
> | + Reasoning CPT | STEM | 71.5 | 74.5 | 56.8 | 71.1 | 67.1 |
> | + Reasoning CPT + SFT | Law | 74.9 | **75.2** | **58.6** | **72.9** | **69.1** |
> | + Reasoning CPT + SFT | STEM | **76.6** | 72.7 | 57.5 | 70.9 | 68.0 |

---

> > ### Author Response · Authors · 2025-12-02
> > **General Response to Common Concerns (3/3)**
> >
> > ## Training with Mixed Domains
> >
> > Similar concerns were raised by the following reviewers:
> > - nb3E (W5): Cross-domain scalability
> > - pwXB (W2 & Q2): Experiments combining Law and STEM data
> > - kkHq (Q2): What happens when Law and STEM are merged?
> >
> > **Response**:
> >
> > Additional Experiments: To address the reviewers' concerns, in addition to models trained on "STEM only" and "Law only," we newly conducted experiments training on a mixed domain combining both domains at 50% each.
> >
> > Effect of Mixed Domains: When trained with 50% mixing, the accuracy was exactly intermediate between training on each domain independently. This trend was consistently observed on both MMLU and GPQA. This suggests that each training data domain has strengths and weaknesses, and mixing averages them out.
> >
> > Improvement in STEM Performance: An interesting point is that for MMLU, 50% mixing showed higher accuracy than training on STEM alone. Training on mixed domain data improves accuracy in Social Sciences and Humanities. This suggests the possibility that mixing domains in Reasoning CPT improves reasoning performance on non-STEM domains beyond STEM alone.
> >
> > We plan to include these explanations in the updated version.
> >
> >
> > **Table 3: MMLU**
> > | Method | Domain | STEM | Social Sciences | Humanities | Other | All |
> > |--------|--------|------|-----------------|------------|-------|-----|
> > | Qwen2.5-7B | - | 69.9 | 74.2 | 57.3 | 70.8 | 66.8 |
> > | + CPT | Law | 70.3 | 73.3 | 56.0 | 71.3 | 66.4 |
> > | + CPT | STEM | 70.4 | 73.4 | 57.0 | 69.8 | 66.5 |
> > | + CPT | STEM–Law 50% mix | 70.9 | 73.8 | 57.3 | 71.1 | 67.0 |
> > | + Reasoning CPT | Law | **72.7** | **75.2** | 57.9 | **71.6** | **68.0** |
> > | + Reasoning CPT | STEM | 71.5 | 74.5 | 56.8 | 71.1 | 67.1 |
> > | + Reasoning CPT | STEM–Law 50% mix | 71.4 | 74.8 | **57.9** | 71.5 | 67.7 |
> >
> > **Table 4: GPQA**
> > | Method | Domain | Biology | Chemistry | Physics | All |
> > |--------|--------|---------|-----------|---------|-----|
> > | Qwen2.5-7B | - | 26.3 | 20.4 | 30.2 | 25.2 |
> > | + CPT | Law | 31.6 | 25.8 | 23.3 | 25.3 |
> > | + CPT | STEM | 31.6 | 24.7 | 24.4 | 25.3 |
> > | + CPT | STEM–Law 50% mix | 31.6 | 25.8 | 24.4 | 25.8 |
> > | + Reasoning CPT | Law | 26.3 | **31.2** | 30.2 | 30.3 |
> > | + Reasoning CPT | STEM | **47.4** | 28.0 | 32.6 | **31.8** |
> > | + Reasoning CPT | STEM–Law 50% mix | 36.8 | 28.0 | **33.7** | 31.3 |

---

### Author Response · Authors · 2025-12-03
**We Thank the reviewers for their valuable feedback**

We sincerely thank all reviewers for your valuable feedback. Multiple reviewers acknowledged: the effectiveness of hidden thoughts (`pwXB`, `kkHq`), the transferability (`pwXB`), novel insights for synthetic data construction (`nb3E`), promising cross-domain transfer results (`kkHq`), and the domain-agnostic approach with thorough experimental support (`mtaB`). Notably, `kkHq` emphasized that "the hidden thoughts on the Law domain seem to really aid the downstream performance in somewhat surprising ways" and praised that "the process is very straightforward" with "good reproducibility," highly evaluating the practical utility of our method.

To address all concerns raised by every reviewer, we have conducted extensive additional experiments and provided clear explanations for missing information. We plan to add these to the updated version of the paper:

- **SFT comparison experiments**: We clarified that Reasoning CPT is orthogonal to existing post-training methods for reasoning models and explained the key advantage of enabling reasoning training from text alone without requiring problem-answer pairs. Moreover, we demonstrated through experiments that Reasoning CPT + SFT achieved the best performance (36.9% on GPQA, 69.1% on MMLU), showing strong complementary effects

- **Domain mixing experiments**:  This experiment revealed the interesting finding that 50% STEM-Law mixing outperformed 100% STEM-only training on MMLU.

- **Subject-level cross-domain analysis**: Training with Hidden Thoughts substantially improves reasoning performance not only on math-derived problems but also across various domains (e.g., STEM-trained model: +12.3% on world_religions, Law-trained Reasoning CPT: +13.0 on college_computer_science,)

- **Updated explanations**: We have clarified the fundamental differences from distillation, CoT, and post-training methods for reasoning models; improved the logical coherence of the abstract and introduction; clarified the definition of hidden thoughts and the purpose of Section 4 analysis; explained the 8× data efficiency advantage of Reasoning CPT over standard CPT; addressed bias concerns; and added missing related work.

Through these extensive additional experiments and detailed explanations, we believe we have successfully addressed all concerns raised by every reviewer.

We are confident that these responses have significantly strengthened our work and are deeply grateful for this opportunity to improve our work.

---

### Meta-Review · Area_Chair_Cgro · 2026-01-04

**Summary:**

The concerns raised by the reviewers are primarily as follows:

1. Comparison with related work and SFT baselines (nb3E, pwXB, mtaB).

2. The presentation of the paper could be improved (nb3E).

3. The domains for evaluation are fairly limited (nb3E, pwXB, kkHq, mtaB; to the domains of law and STEM), and a lot of conclusions are made around these two domains, which raise questions about the generalizability of the findings.

**Reviewer Concerns:**

The authors have worked hard to improve the paper. Concern 1 has been adequately addressed in the rebuttal, but the authors claim they addressed the concerns by training on a mixture of domains, which does not fully address the point.

**Reviewer Scores:**

I don't think Reviewer kkHq will change their rating. However, other three reviewers might raise their ratings for the many experiments newly added by the authors.

---

### Decision · Program_Chairs · 2026-01-26

Reject